# Using automated machine learning for the upscaling of gross primary productivity

Max Gaber[1,2], Yanghui Kang[1,3], Guy Schurgers[2], Trevor Keenan[1,3]

[1] Department of Environmental Science, Policy and Management, UC Berkeley, Berkeley, CA 94720, USA.
[2] Department of Geosciences and Natural Resource Management, University of Copenhagen, Copenhagen, 1350, Denmark.
[3] Climate and Ecosystem Sciences Division, Lawrence Berkeley National Laboratory, Berkeley, CA 94720, USA.

*Correspondence to*: Max Gaber (mfg@ign.ku.dk), Yanghui Kang (yanghuikang@berkeley.edu), Trevor Keenan (trevorkeenan@berkeley.edu)

**Abstract.**

Estimating gross primary productivity (GPP) over space and time is fundamental for understanding the response of the terrestrial biosphere to climate change. Eddy covariance flux towers provide *in situ* estimates of GPP at the ecosystem scale, but their sparse geographical distribution limits larger-scale inference. Machine learning (ML) techniques have been used to address this problem by extrapolating local GPP measurements over space using satellite remote sensing data. However, the accuracy of the regression model can be affected by uncertainties introduced by model selection, parameterization, and choice

of explanatory features, among others. Recent advances in automated ML (AutoML) provide a novel automated way to select and synthesize different ML models. In this work, we explore the potential of AutoML by training three major AutoML frameworks on eddy-covariance measurements of GPP at 243 globally distributed sites. We compared their ability to predict GPP and its spatial and temporal variability based on different sets of remote sensing explanatory variables. Explanatory variables from only MODIS surface reflectance data and photosynthetically active radiation explained over 70 % of the

monthly variability in GPP, while satellite-derived proxies for canopy structure, photosynthetic activity, environmental stressors, and meteorological variables from reanalysis (ERA5-Land) further improved the frameworks' predictive ability. We found that the AutoML framework AutoSklearn consistently outperformed other AutoML frameworks as well as a classical Random Forest regressor in predicting GPP, but with small performance differences, reaching an $r^2$ of up to 0.75. We deployed the best-performing framework to generate global wall-to-wall maps highlighting GPP patterns in good agreement with

satellite-derived reference data. This research benchmarks the application of AutoML in GPP estimation and assesses its potential and limitations in quantifying global photosynthetic activity.

## 1 Introduction

Terrestrial gross primary productivity (GPP) describes the gross photosynthetic assimilation of atmospheric carbon dioxide ($CO_2$) at the ecosystem scale. As the largest flux in the global carbon cycle, GPP plays a vital role in maintaining ecosystem

functions and sustaining human well-being (Beer et al., 2010; Friedlingstein et al., 2019). In addition, the dynamics of GPP directly affect the growth rate of atmospheric $CO_2$ concentrations and ecosystem feedbacks to the climate system. Therefore, accurate estimates of the magnitude and spatiotemporal patterns of terrestrial GPP are essential for understanding ecosystem carbon cycling and developing effective climate change mitigation and adaptation strategies (Keenan et al., 2016; Canadell et al., 2021).


While *in situ* GPP estimates are available from methods such as the eddy covariance technique, global spatiotemporal patterns are challenging to estimate due to the lack of large-scale observations and the high uncertainty of process-based vegetation models (Anav et al., 2015). Fluxes captured by the eddy covariance measurements are limited to the area within the tower's footprint, typically ranging from several hundred meters to several kilometers (Gong et al., 2009). Therefore, various data-

driven methods such as machine learning (ML) have been used to scale up *in situ* GPP measurements from flux tower networks to a global scale. These ML models use independent globally available explanatory data from remote sensing or other continuous model outputs to infer a functional relationship to the GPP measurements, which can be used to predict GPP in areas beyond the limited flux tower footprints. Commonly applied models include tree-based methods (Bodesheim et al., 2018; Wei et al., 2017; Beer et al., 2010; Jung et al., 2011), artificial neural networks (Joiner and Yoshida, 2020; Beer et al., 2010;

Papale et al., 2015), linear regressors, kernel methods, and ensembles thereof (Tramontana et al., 2016). Despite the wide variety of ML models applied, a high degree of uncertainty remains in the selection of appropriate features, algorithms, and configurations (Reichstein et al., 2019). The data-based models typically perform well in estimating seasonal GPP patterns but show limitations in predicting trends and interannual variability (Tramontana et al., 2016).

The contribution of different explanatory variables, such as greenness measures, photosynthetically active radiation (PAR), land surface temperature (LST), soil moisture (SM), and meteorological variables (vapor pressure deficit, temperature, precipitation) to the accuracy of the GPP predictions (hereafter referred to as variable importance) has not been conclusively clarified. Both Tramontana et al. (2016) and Joiner and Yoshida (2020) confirmed the dominant control of remotely sensed greenness on the ML prediction of GPP at daily to interannual time scales, with meteorological variables contributing

marginally. Conversely, Stocker et al. (2018) found an important control of site-measured soil moisture on light use efficiency (LUE) and GPP at daily granularity under drought conditions at flux sites. Furthermore, Dannenberg et al. (2023) showed that including satellite-derived soil moisture and LST data significantly improved the estimation of monthly GPP in drylands over the western US. However, a comprehensive assessment of the importance of meteorological and satellite-derived variables beyond vegetation structure at the global scale is lacking. Given the ubiquitous intercorrelation among remote sensing and

meteorological variables, the importance of different explanatory variables has typically been accomplished by training separate models on different input combinations (Tramontana et al., 2016). Yet, ML model performance can vary strongly depending on the dimension of input features, hyperparameter tuning (the search for the optimal parameters that control the learning process of an ML model), and even the specific type of ML model employed (Raschka, 2020; Cawley and Talbot, 2010). Therefore, a unified ML framework that concurrently optimizes model choice and parameterization is required to

facilitate a balanced assessment of driver importance in global GPP upscaling.

Navigating the search space created by the choice of model architecture, hyperparameters, and preprocessing steps to find a suitable combination for GPP prediction is a resource-intensive task. Therefore, researchers often evaluate a selection of combinations that they expect to perform well, thereby potentially missing out on the optimal solution (Karmaker et al., 2021).

Automated machine learning (AutoML) aims to overcome these challenges through an autonomous approach. By evaluating different combinations of preprocessing steps, candidate ML models, and hyperparameters, AutoML aims to find the optimal ML configuration for the given ML problem and available training data. In addition, it leverages the unique strengths of different algorithms by using ensembling or stacking techniques. At the time of this study, AutoML is still under ongoing development but has recently received increasing attention in the environmental sciences and beyond. It has shown superior

performance to classical ML, for example, in modeling water nutrient concentrations (Kim et al., 2020), dam water inflows (Lee et al., 2023), and water quality prediction (Madni et al., 2023), and similar performance to reference models for climate zone classification (Traoré et al., 2021) and drought forecasts (Duan and Zhang, 2022). Other use cases include predicting landslide hazards (Qi et al., 2021), root zone soil moisture (Babaeian et al., 2021), or GPP at a single flux tower site (Guevara-Escobar et al., 2021).


In this study, we investigate if and how AutoML can improve global GPP upscaling at the monthly frequency from *in situ* measurements using globally available explanatory variables. We examine the three frameworks AutoSklearn, H2O AutoML,

and AutoGluon in this study since they have shown outstanding performance in benchmarks and Kaggle competitions (Guyon et al., 2019; Erickson et al., 2020; Truong et al., 2019; LeDell and Poirier, 2020; Feurer et al., 2018). All frameworks differ in
their architecture and approach to selecting ML algorithms. We evaluate their selection of processing and ML algorithms based on site-level measurements. In addition, we evaluate the variable importance, i.e., the contribution of various remotely sensed vegetation structure variables, proxies for photosynthetic activity and environmental stress (i.e., greenness, land surface temperature, soil moisture, evapotranspiration), and meteorological factors to the performance of the AutoML frameworks. The impacts of the spatial resolution of remote sensing data on GPP estimation are further assessed. Finally, we upscale our
results to global wall-to-wall GPP maps and evaluate their spatio-temporal patterns and associated uncertainties.

## 2 Methods and materials

### 2.1 Data

#### 2.1.1 Eddy covariance measurements

We merged eddy covariance datasets from FLUXNET 2015 (Pastorello et al., 2020), AmeriFlux FLUXNET
(https://ameriflux.lbl.gov/data/flux-data-products), and ICOS Warm Winter 2020 (ICOS, 2020) to obtain a large number of monthly GPP estimates from net ecosystem exchange (NEE) measurements. Where sites were available in more than one source, we kept the most recent record. The data quality control followed previous studies (Tramontana et al., 2016; Jung et al., 2011; Joiner et al., 2018). We considered monthly values where at least 80% of the NEE data came from actual measurements or were high-quality gap-filled. We used the GPP derived from NEE using the night-time partitioning approach
(Reichstein et al., 2005), and negative GPP outliers were truncated at -1 gC $m^{-2}$ $d^{-1}$ average daily GPP.

The preprocessing resulted in a dataset of 243 sites and 18,218 site-months, ranging from 2001 to 2020, and serving as the ground truth for the evaluation of site-level GPP predictions (Fig. 1). The distribution of sites and site-months shows strong biases in region, biome, and climate representation (Fig. 2). We reorganized the land cover classes, as individual land cover
classes related to shrublands and savannas rarely occurred. Therefore, "open shrublands" and "closed shrublands" were merged, as well as "savannas" and "woody savannas", resulting in the following land cover according to the International Geosphere–Biosphere Programme (IGBP)(International Geosphere–Biosphere Programme, 2024): croplands (CRO), shrublands (SH), deciduous broadleaf forests (DBF), evergreen broadleaf forests (EBF), evergreen needleleaf forests (ENF), grasslands (GRA), mixed forests (MF), savannas (SAV), permanent wetlands (WET), and the non-vegetated classes of
permanent snow and ice (SNO), water bodies (WAT), and barren soil (BAR).

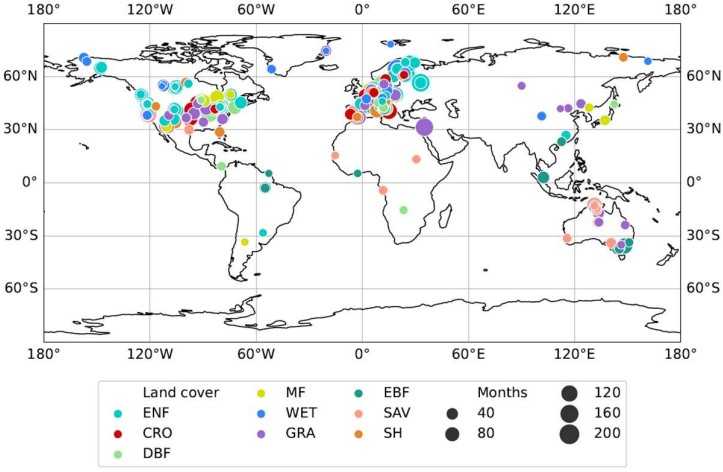

**Figure 1: Locations of the measurement sites. The marker size represents the number of monthly measurements available at the respective location. The color stands for the land cover class reported at the site and comprises croplands (CRO), shrublands (SH),**

deciduous broadleaf forests (DBF), evergreen broadleaf forests (EBF), evergreen needleleaf forests (ENF), grasslands (GRA), mixed
forests (MF), savannas (SAV), and permanent wetlands (WET).

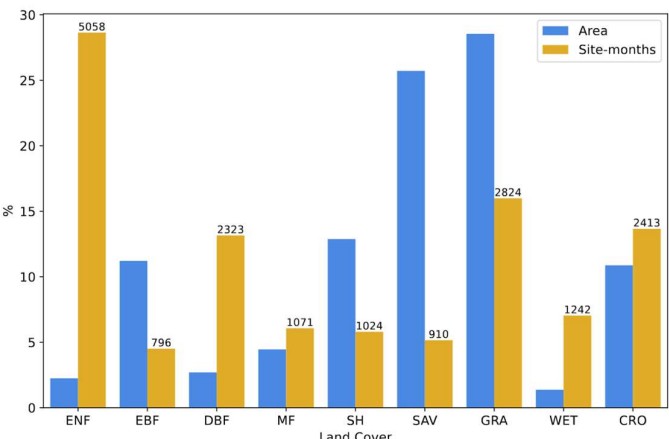

Figure 2: Standardized number of sites-months and global area of each land cover type, excluding land covers without any GPP
measurements. The number of sites-months is shown above their respective columns. The land cover classes reported follow the
IGBP classification (International Geosphere–Biosphere Programme, 2024) and comprise croplands (CRO), shrublands (SH),
deciduous broadleaf forests (DBF), evergreen broadleaf forests (EBF), evergreen needleleaf forests (ENF), grasslands (GRA), mixed
forests (MF), savannas (SAV), and permanent wetlands (WET).

**2.1.2 Explanatory variables**

Our goal was to provide as many explanatory variables as possible and let the frameworks decide which to use. We obtained

gridded explanatory variables from various sources of remotely sensed and modeled data with global coverage. The data

allowed us to evaluate locally by sampling at the tower locations and to predict on a global wall-to-wall scale. These variables

include products based on Moderate Resolution Imaging Spectroradiometer (MODIS) measurements, such as nadir-BRDF

adjusted reflectances (NBAR) from optical to infrared wavelengths, the fraction of photosynthetically active radiation (FPAR),

leaf area index (LAI), day and night surface temperature, and land cover. We also included the photosynthetically active

radiation (PAR), diffuse PAR, and the surface downwelling shortwave flux (RSDN) from BESS_Rad, as well as solar-induced

fluorescence (SIF), evapotranspiration (ET), and soil moisture (SM). In addition, we used meteorological data from the ERA5-

Land reanalysis, including precipitation, temperature, and vapor pressure deficit (VPD). We applied a three-month lag in

precipitation to account for water availability. Table 1 shows an overview of all explanatory variables.

Many of the explanatory variables are themselves datasets that have been modeled from MODIS data. For instance, SIF was

predicted from MODIS NBAR using a feed-forward neural network trained on OCO-2 SIF retrievals (Zhang et al., 2018). ET

estimates were modeled by a coupled land-surface and atmospheric boundary layer model (Atmosphere Land Exchange

Inverse, ALEXI), which used MODIS LST and LAI as inputs, among others (Hain and Anderson, 2017). Although their input

data largely overlap with the inputs to our model, we expected additional improvements from including these datasets due to

the domain knowledge of their models, which would otherwise be difficult to replicate in this study by solely relying on

MODIS data and limited GPP measurements.

We filtered the data for poor-quality pixels, performed gap-filling, and matched spatial and temporal resolutions. We used

NBAR (MCD43C4 v006), where more than 75 % of high-resolution NBAR pixels were available from the full BRDF

inversion. We selected LST data by applying the quality control mask, and where the average emissivity error was less than

0.02. LAI and FPAR were used when retrieved using the main algorithm with or without saturation. Data gaps were filled at

the native resolution, similar to the procedure of Walther et al. (2022). We filled gaps of less or equal five days (8 days for

four-day resolution datasets) with the average of a fifteen-days moving window for high-frequency datasets (NBAR, LAI,

FPAR, BESS_Rad, CSIF). We gap-filled LST with a 9-day moving window because we observed higher variations. For SM,

we used the moving window median for short gaps and the mean seasonal cycle for long gaps. Finally, we resampled all datasets to 0.05 ° spatial resolution and monthly temporal resolution. Coarser-resolution datasets were resampled using a nearest-neighbor approach, while high-resolution data was down-sampled using the conservative remapping method (Jones, 1999).

**Table 1: Explanatory variables and sources and their respective spatial and temporal resolution.**

| Explanatory Variable | Source | Spatial Resolution | Temporal Resolution |
|---|---|---|---|
| Reflectance (Nadir-BRDF adjusted; NBAR) Bands 1–7 | MODIS MCD43C4 v006 (Schaaf and Wang, 2015) | 0.05 ° | daily |
| PAR | BESS_Rad (Ryu et al., 2018) | 0.05 ° | daily |
| Diffuse PAR | BESS_Rad (Ryu et al., 2018) | 0.05 ° | daily |
| RSDN | BESS_Rad (Ryu et al., 2018) | 0.05 ° | daily |
| FPAR | MODIS MCD15A2H v006 (Myneni et al., 2015) | 500m | 4 days |
| LAI | MODIS MCD15A2H v006 (Myneni et al., 2015) | 500m | 4 days |
| Land surface temperature (day) | MODIS MYD11A1, MOD11A1 (Wan et al., 2015) | 1km | daily |
| Land surface temperature (night) | MODIS MYD11A1, MOD11A1 (Wan et al., 2015) | 1km | daily |
| Evapotranspiration | ALEXI (Hain and Anderson, 2017) | 0.05 ° | daily |
| Soil moisture | ESA CCI v.06.1 (Gruber et al., 2019) | 0.25 ° | daily |
| SIF | CSIF (Zhang et al., 2018) | 0.05 ° | 4 days |
| Instantaneous SIF | CSIF (Zhang et al., 2018) | 0.05 ° | 4 days |
| Land cover (biome) | MODIS MCD12Q1 (Friedl and Sulla-Menashe, 2019) | 500m | annual |
| Total precipitation | ERA5-Land (Muñoz-Sabater et al., 2021) | 0.1 ° | hourly |
| Total precipitation (3 months lag) | ERA5-Land (Muñoz-Sabater et al., 2021) | 0.1 ° | hourly |
| Temperature | ERA5-Land (Muñoz-Sabater et al., 2021) | 0.1 ° | hourly |
| Vapor Pressure Deficit (VPD) | ERA5-Land (Muñoz-Sabater et al., 2021) | 0.1 ° | hourly |

**2.2 Automated machine learning**

The performance of ML is highly dependent on the selection and configuration of preprocessing steps, model architectures, and corresponding hyperparameters, which are determined by the specific ML problem (Hutter et al., 2019). The steps involved are typically organized sequentially in an ML pipeline and transform the input features (explanatory variables) into a target variable (Zöller and Huber, 2021). The pipeline refers to the entire process of developing and training an ML model and typically consists of several tasks, such as preprocessing, feature engineering, model training, hyperparameter tuning, and model deployment.

Selecting the appropriate algorithms and hyperparameters is often referred to as the combined algorithm selection and hyperparameter tuning (CASH) problem and involves exploiting a search space spanned by the available algorithms and their parameters. Solving the CASH problem is challenging because the search space is high-dimensional and hierarchical, and its exhaustive exploitation is often computationally expensive (Kotthoff et al., 2019; Thornton et al., 2013). As a result, candidate pipeline configurations are typically determined in controlled experiments using optimization methods, such as grid search, randomized search, and Bayesian optimization, or through experience and educated guesswork (Karmaker et al., 2021).

In contrast, AutoML provides an optimization approach with an end-to-end scope. A fully developed AutoML framework iteratively selects the pipeline structure, algorithms, and hyperparameters from the search space based on data requirements and objective functions while considering a time and resource budget (Yao et al., 2019). Thus, it facilitates usability for domain experts and overcomes inefficient trial-and-error approaches. AutoML draws from a pool of classical ML algorithms (base models) and preprocessing methods and selects or combines the most appropriate candidates for the ML problem. Typically, AutoML frameworks create model ensembles by combining the predictions of their base models, either through a simple aggregation or through yet another model that uses the predictions of the base models as input features. This approach is often superior to individual predictions because it can overcome the limitations of the individual base models (van der Laan et al., 2007).

AutoML frameworks handle pipeline creation with various degrees of autonomy and scope, given the early-stage development of much of the available software at the time of this study. For example, tasks such as pipeline selection or feature engineering are only sporadically implemented in the available frameworks (Zöller and Huber, 2021). With H2O AutoML, AutoSklearn, and AutoGluon, we compared AutoML frameworks that differ in training procedure, optimization method, and available base models, and have been tested in a wide range of applications and benchmarks (Balaji and Allen, 2018; Truong et al., 2019; Erickson et al., 2020; Hanussek et al., 2020; Ferreira et al., 2021).

**AutoSklearn**

AutoSklearn (Feurer et al., 2015) is an AutoML library built on top of the Scikit-Learn ML models. We used AutoSklearn in version 0.14.7. The framework relies on a wide range of base models, including AdaBoost, ARD regression, Decision Trees, Extra Trees, Gaussian processes, Gradient Boosting, k-Nearest Neighbors, Support Vector regression, MLP regression, Random Forests, and SGD regression. It also considers feature engineering algorithms, such as PCA, percentile regression, and feature agglomeration (AutoSklearn, n.d.). The framework selects and tunes its base models in a Bayesian optimization and performs a forward stepwise ensemble selection (Caruana et al., 2004). During this process, the framework draws on a pool of ML models to build the model ensemble, but instead of using the entire pool, it adds the models one by one, only using the ones that maximize ensemble performance. AutoSklean also uses a meta-learner trained on the meta-features of a variety of datasets to warm start the optimization procedure, which increases efficiency and reduces training time (Feurer et al., 2015). The meta-learner uses knowledge from previous experiments with similar datasets and can, therefore, select promising ML models to start with instead of training from scratch each time.

**H2O AutoML**

H2O AutoML (LeDell and Poirier, 2020) is a widely used AutoML framework for supervised regression and classification. We used H2O 3 and the Python package of version 3.18.0.2. H2O AutoML draws from a set of base models, which, in the developer's terminology, are divided into the model families of Gradient Boosting models (GBM), XGBoost GBMs, GLMs, a default Random Forest model (DRF), Extremely Randomized Trees (XRT), and feed-forward neural networks. The framework trains these models in a predefined order with increasing diversity and complexity, using pre-specified

hyperparameters or tuning them by random search. In addition to the individual base models, H2O AutoML creates ensembles of the base models, combining their predictions through a generalized linear model (GLM) by default. The ensembles consist of either all base models or only the best-performing base models from each model family. H2O AutoML then ranks the performance of individual models and model ensembles using an internal cross-validation (CV). The best-performing model is used for prediction.

**AutoGluon**

AutoGluon Tabular (Erickson et al., 2020) relies heavily on ensemble and stacking techniques. It differs from many other frameworks by omitting model selection and hyperparameter tuning, thus avoiding the computationally intensive CASH problem. The framework draws from a pool of base models: neural networks, LightGBM boosted trees, Random Forests, Extremely Randomized Trees, and k-Nearest Neighbors. These models are combined in a multi-layer stack ensembling process: AutoGluon first generates predictions from each base model. The predictions are then concatenated with the original features and passed to another set of models (the stacker models) in the next layer. Their predictions can be concatenated again and passed to the next layer, and so on, creating a layered structure of model sets and concatenation steps. The predictions of the last layer are combined in an ensemble selection step (Caruana et al., 2004). Each layer consists of the same base model types and hyperparameters. In addition, AutoGluon implements *k*-fold bagging, which improves performance by using the training data more efficiently. Global and model-specific preprocessing algorithms are available to impute missing values or correct skewed distributions. A feature selection algorithm is provided in the framework but is still in an experimental stage and not enabled in the version used.

**2.3 Experimental design**

We first evaluated the three AutoML frameworks under three sets of explanatory variables. In addition, we trained a classical Random Forest model in a randomized search, which served as our baseline. We then used AutoSklearn with the best-performing set of explanatory variables to upscale *in situ* eddy covariance GPP measurements to global wall-to-wall maps (Fig. 3).

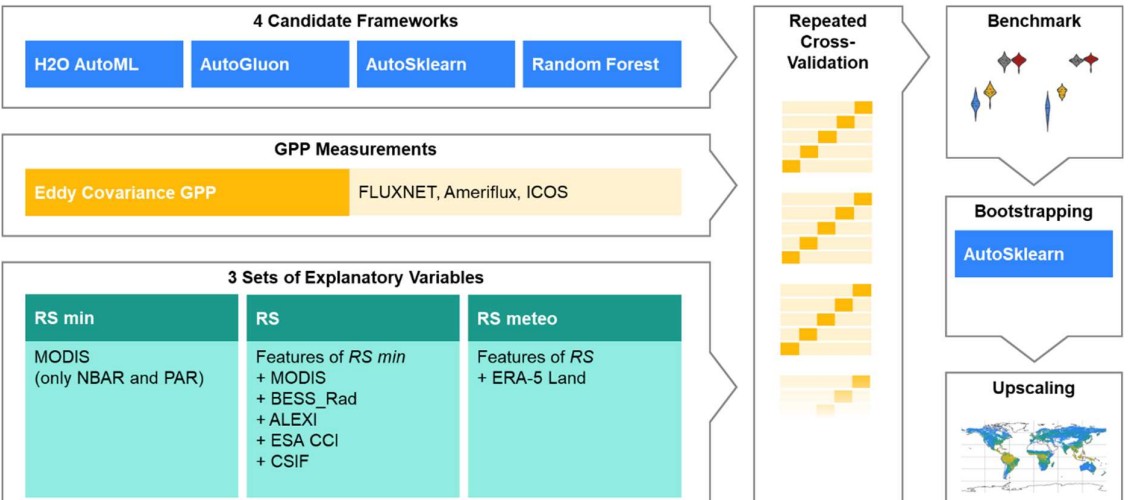

**Figure 3 Experiment setup. We trained and evaluated AutoSklearn, H2O AutoML, AutoGluon, and Random Forest, together with three sets of explanatory variables in repeated cross-validation on GPP data from eddy covariance measurements. Then, we trained AutoSklearn in a bootstrap aggregation to produce global wall-to-wall GPP maps. The abbreviations of the explanatory variable sets translate as follows: "RS" for remotely sensed and "meteo" for meteorological data.**

### 2.3.1 Explanatory variable sets

We organized the explanatory variables into three sets to determine their impact on GPP predictions within different AutoML frameworks (Tramontana et al., 2016; Joiner and Yoshida, 2020). Each set consisted of different features that could explain the variation in GPP. The minimal set of remotely sensed variables (RS minimal) included surface reflectance from seven MODIS visible to infrared bands and PAR, which largely reflect the ability of the vegetation canopy to intercept solar radiation for photosynthesis. The "RS" set included all remotely sensed variables and their products. Notably, compared to the "RS minimal" set, the "RS" set also included land surface temperature, evapotranspiration, and soil moisture, which provide an additional link to vegetation heat and water stress (Green et al., 2022; Stocker et al., 2018). Finally, the "RS meteo" set included all remotely sensed variables and, in addition, meteorological variables from the ERA5-Land reanalysis (see Table 2). Additionally, we replaced the MODIS reflectance bands, LAI, FPAR, and land cover products with their native 500 m resolution data in the "RS" set to evaluate the impact of satellite data spatial resolution on GPP estimation.

**Table 2 Explanatory variable sets and associated data sets.**

| Explanatory Variable | RS minimal | RS | RS meteo |
|---|:---:|:---:|:---:|
| Reflectance (Nadir-BRDF adjusted; NBAR), Bands 1–7 | ● | ● | ● |
| PAR | ● | ● | ● |
| Diffuse PAR | | ● | ● |
| RSDN | | ● | ● |
| FPAR | | ● | ● |
| LAI | | ● | ● |
| Land surface temperature (day) | | ● | ● |
| Land surface temperature (night) | | ● | ● |
| ET | | ● | ● |
| Soil moisture | | ● | ● |
| SIF | | ● | ● |
| Instantaneous SIF | | ● | ● |
| Land cover (biome) | | ● | ● |
| Total precipitation | | | ● |
| Total precipitation (3 months lag) | | | ● |
| Temperature | | | ● |
| Vapor Pressure Deficit (VPD) | | | ● |

The explanatory variable sets can provide information about the importance of the input features on the performance of the upscaling frameworks. They are particularly important as many of the AutoML frameworks lack feature engineering algorithms and cannot select relevant features themselves.

### 2.3.2 Framework assessment

We used five-fold cross-validation to train and evaluate the AutoML frameworks. Grouping the data by site helped us increase the independence between the folds and evaluate the models' ability to generalize spatially. Thus, a time series at one site could be assigned to only one fold and not split into training and test sets. In addition, stratification by land cover helped to distribute the folds similarly. We repeated the cross-validation thirty times with different random splits to evaluate the impact of partitioning the data on the final performance in our evaluation.

With H2O AutoML, AutoSklearn, and AutoGluon, we selected popular frameworks for supervised regression problems on tabular data that support parallelization and a Python interface. Since AutoML is intended to work as an out-of-the-box solution, we kept the frameworks' configurations at default or recommended parameter values where it was possible and reasonable to do so. Moreover, we set each framework to optimize for the root mean squared error (RMSE) and limited the resource usage during training to 600 CPU minutes per CV fold (30 minutes on 20 CPUs) and 64GB of memory.

We used the RMSE and the coefficient of determination ($r^2$) to evaluate the frameworks' performance by comparing the out-of-fold predictions to the ground truth values of GPP (Eq. A1). The latter aligns with the Nash-Sutcliffe model efficiency (Nash and Sutcliffe, 1970) used in some literature as a performance metric for the GPP prediction (e.g., Tramontana et al. (2016)). In addition to obtaining performance metrics for the total time series prediction, we decomposed the time series to evaluate the performance in different spatial and temporal domains. We computed the components as follows: we obtained trends by linear regression of the entire time series (using the slope for evaluation with RMSE and $r^2$), seasonality (mean seasonal cycle) by month-wise averaging, and anomalies as their residuals after detrending and removing seasonality. Furthermore, we calculated an across-site variability from the multi-year mean at each site. For this analysis, we considered only sites with a minimum of 24 months of measurements to minimize the error from sites with just a few measurements, leaving us with 211 sites. When calculating trend metrics, we only considered sites with at least 60 months of measurements for our trend evaluations. Time series anomalies were detrended only when this minimum was reached; otherwise, we simply removed the seasonal component from the time series.

Moreover, we tested how the average ranked performance of each framework compared to the other frameworks. We calculated the performance ranks within each repeated cross-validation and obtained an average rank for each framework. Using the Friedman test, we tested for statistically significant differences in the rank distribution, evaluating the null hypothesis of no significant differences with a significance level of 0.01. We then used the Nemenyi post hoc test to find frameworks with significant differences in mean rank while adjusting for type I error inflation by using a family-wise error correction. We rejected the null hypothesis (no significant difference between the two frameworks) if the difference between the average ranks exceeded a critical difference (CD), which depends on the critical value of the Studentized range distribution (Demšar, 2006).

### 2.3.3 GPP upscaling

We used AutoSklearn with the "RS" explanatory variable set to upscale the eddy covariance measurements to a global scale, as this combination of framework and explanatory variables performed best in the benchmark. We trained thirty models in a bootstrap aggregation approach, where each bootstrap was sampled with replacement to a size of 80 % of the total number of sites. We kept the time series grouped by site but removed the land cover stratification. This technique allowed us to estimate GPP as the mean of the bootstrapped predictions and provided a sampling error (standard error of the mean) as a spatially distributed uncertainty estimate for the model prediction. We produced global GPP and standard error maps at a resolution of 0.05 ° in monthly frequency from 2001 to 2020, which we compared with the two ML-based reference datasets FluxCom v6 (RS only, based on data from the MODIS collection 6) (Jung et al., 2020) and FluxSat (Joiner and Yoshida, 2020).

## 3 Results

### 3.1 AutoML Framework performance

In general, we found that all frameworks perform in a close range of coefficients of determination ($r^2$), explaining on average between 70% and 75% of the variation in eddy covariance GPP measurements. However, the performance depends on the

framework used and the selection of variables. Examining the distribution of r²values for the different repeated cross-validations, we can see that AutoSklearn performs best, followed by H2O AutoML, Random Forest, and AutoGluon in predicting monthly GPP (Fig. 4). AutoSklearn achieved the highest $r^2$ among the four frameworks for all explanatory feature sets. A similar pattern is observed for trends, seasonality, across-site variability, and anomalies (Fig. 5). Note that we removed one outlier for H2O AutoML trained on the "RS" variable set, which deviated more than five standard deviations from the mean value due to very low performance in one CV fold.

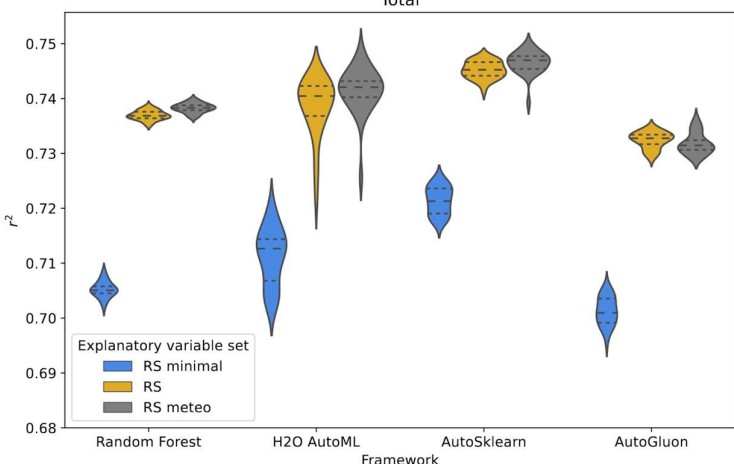

**Figure 4 Overall framework performance, expressed as the coefficient of determination (r²) for the candidate frameworks and the three different explanatory variable sets. Each distribution belongs to one framework and one set of explanatory variables and results from the repeated cross-validations, for each of which one r² value is calculated over the predictions at all sites.**

AutoSklearn's superior performance is primarily due to its ability to capture seasonal components and across-site variability (Fig. 5). When trained on "RS" explanatory variables, AutoSklearn achieved average $r^2$ values of $0.7452 \pm 0.0003$ overall, and $0.483 \pm 0.002$ for trends, $0.8142 \pm 0.0003$ for seasonalities, and $0.689 \pm 0.001$ for across-site variability. However, all models struggle to reproduce the monthly anomalies, explaining less than 11 % of the variability (AutoSklearn: $10.40 \pm 0.04$ %). Uncertainties are reported as the standard error of the mean of all cross-validation results.

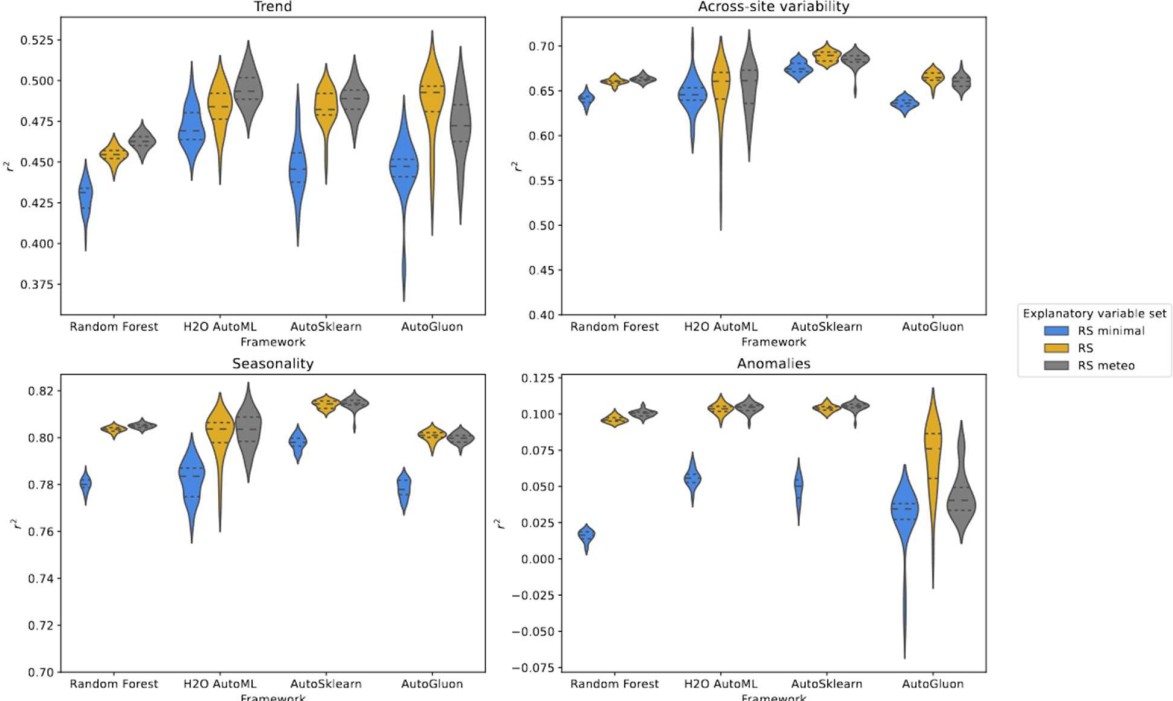

**Figure 5 Evaluation of the temporally and spatially decomposed time series expressed as the coefficient of determination ($r^2$). Each distribution belongs to one framework and one set of explanatory variables and results from the repeated cross-validations, for each of which one r2 value is calculated over the predictions at all sites. The $r^2$ values for seasonality and anomalies were calculated from seasonal cycles and anomalies at monthly granularity, while those for trend and across-site variability were calculated from one trend or mean value per site, respectively.**

Using the Friedman test, we found that the four ML frameworks are statistically different in their performance in predicting monthly GPP as well as its trends, seasonality, anomaly, and across-site variability (p-value < 0.01). However, their difference in performance is marginal. The Nemenyi post hoc test shows that for the "RS" explanatory variables, AutoSklearn achieves the highest average rank with statistical significance among all frameworks for monthly GPP and all its components (Fig. 6a). For the prediction of anomalies, we could not find a significant difference in the average rank between AutoSklearn and H2O AutoML. Trends were predicted by all AutoML frameworks without significant differences in rank. Random Forest and AutoGluon perform the worst, while they are not statistically different in predicting across-site variability and seasonalities.

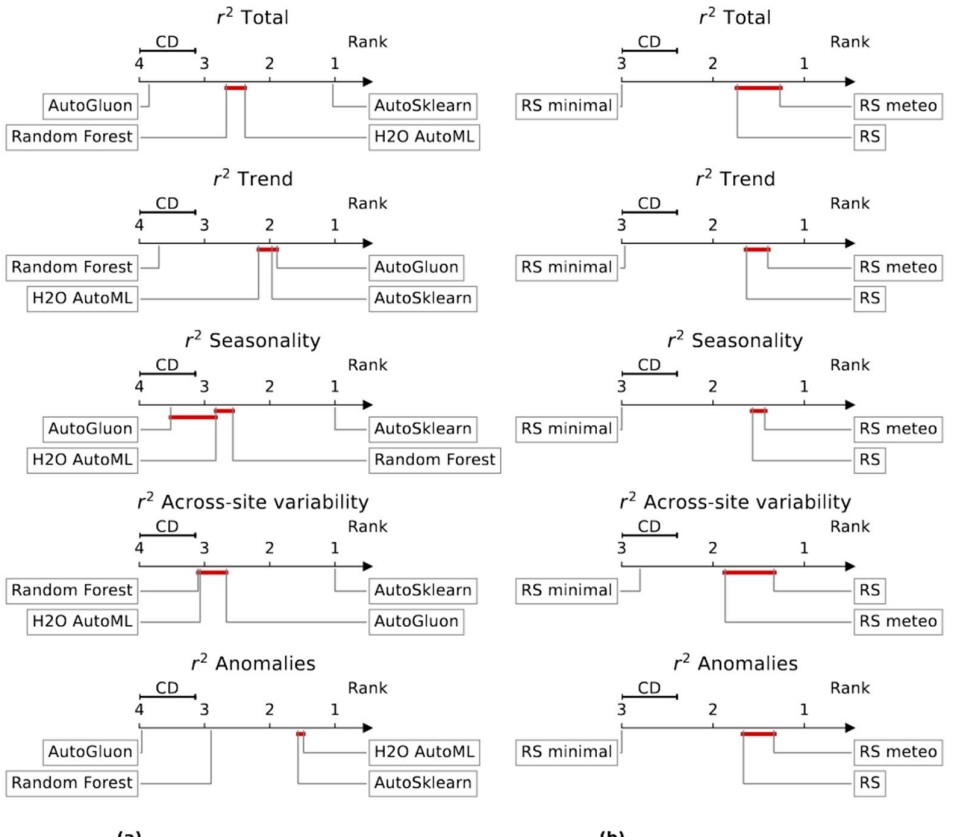

(a)                          (b)

**Figure 6 Critical difference (CD) diagrams (Demšar, 2006) for the ranks of the frameworks and variable sets, which are typically used to compare the performance of multiple algorithms on multiple problems (in this case, repeated cross-validations). The graphs rank the performance of different framework-variable combinations on the x-axis, with one being the best rank. The ranks shown are the average ranks from all repeated cross-validations for each of the frameworks/variable sets. The performance ($r^2$) is given for**
**predicting total GPP and for its different spatial and temporal components: trend, seasonality, anomalies, and across-site variability. We evaluated whether the ranks are statistically significantly different from each other using the critical difference (CD) obtained from a Nemenyi post hoc test. If the difference between the ranks is less than the CD, we assume a nonsignificant difference in ranks, indicated by a red crossbar between the rank markers. On the left side (a), the ranks of the frameworks trained on the "RS" explanatory variables are shown. On the right side (b), the ranks of AutoSklearn trained on different sets of explanatory variables**
**are shown.**

The selection of explanatory variables had a significant impact on the performance of the frameworks. Models with only surface reflectance and PAR (RS minimal) explained the least amount of GPP variability (70–72 %) (Fig. 4). The greatest improvement occurred with the "RS" set when information on SIF, FPAR, LAI, LST, ET, soil moisture, and biome type was included. The "RS" set increased $r^2$ on "RS minimal " by about 0.02 for all frameworks, with sizable improvements in
predicting trends and anomalies (Fig. 5). Meteorological variables slightly improved the prediction of monthly GPP by better explaining spatial variability, trends, and anomalies except for AutoGluon (Fig. 5). However, statistical tests of model ranks showed no significant advantage in the rank of the "RS meteo" over the "RS" set of explanatory variables in any of the decomposed time series features and frameworks (Fig. 6b). The "RS" set outperformed "RS minimal" for predicting GPP and all of its spatiotemporal components. Except for the performance of Random Forest on across-site variability, trend, and
anomalies, "RS" was always the best-performing variable set or insignificantly different from the best-performing variable set. In addition, we evaluated whether vegetation indices (VI) could improve the performance of the variable sets, but no improvements were found beyond the "RS minimal" dataset (Tab. A1).

To determine which explanatory variable was most effective for predicting GPP, we evaluated the permutation importance of
350 the variables for the AutoSklearn framework. Permutation importance is the decrease in prediction performance on the test dataset when one of the variables is randomly shuffled to break its relationship with the target variable. To deal with collinearity

among the explanatory variables (Fig. A1), we first clustered them based on their average mutual Pearson correlation coefficient, regardless of their data source or ecological function. Variables with an average correlation greater than 0.7 were clustered and permuted together, resulting in clusters focused around specific meteorological characteristics (e.g., precipitation, temperature), vegetation properties, or combinations of reflectance bands but also combining features that are not directly biophysically related (Fig. A2 and A3).

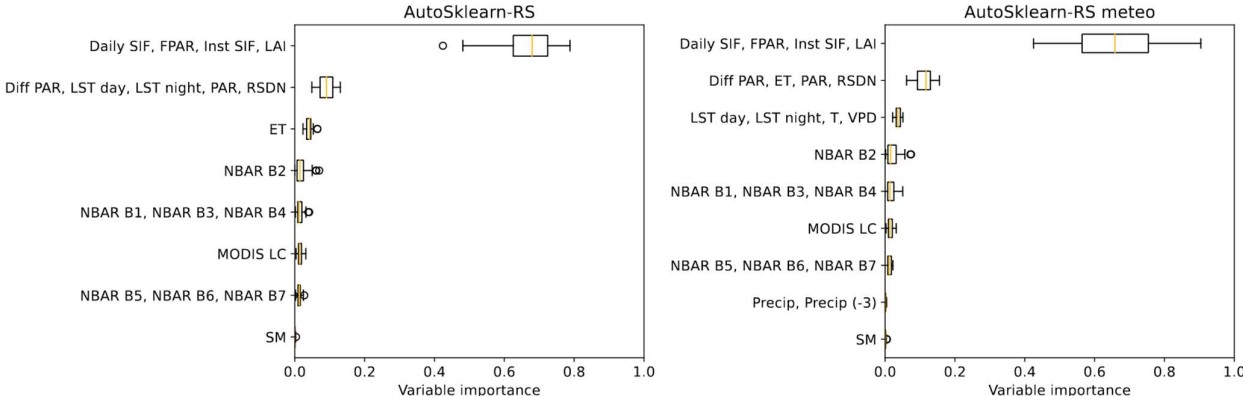

**Figure 7 Permutation importance for different explanatory variables with the AutoSklearn framework and "RS" and "RS meteo" variable sets. The variables are grouped into clusters of colinear variables regardless of data source or ecological function. The importance is the decrease of r² at test time when the variables of the corresponding cluster are randomly shuffled. The variables include the MODIS NBAR bands (red, NIR, blue, green, and 3 SWIR bands), land surface temperature (LST), leaf area index (LAI), photosynthetically active radiation (PAR), fraction of absorbed PAR (FPAR), diffuse PAR (Diff PAR), daily and instantaneous solar-induced fluorescence (SIF), surface downwelling shortwave flux (RSDN), soil moisture (SM), evapotranspiration (ET), precipitation (Precip), temperature at 2m height (T), vapor pressure deficit (VPD), and precipitation with 3-months lag (Precip (-3)). The distribution results from the repeated cross-validations, for each of which one r² value is calculated over the predictions at all sites.**

Our results show the largest decrease in r² of AutoSklean-RS when removing the cluster of SIF, LAI, and FPAR, followed by PAR, RSDN, LST, and ET (Fig. 7). The other variables do not substantially reduce the framework performance. Trained on "RS meteo," AutoSklearn's variable importance gives a similar picture despite slightly different clusters due to the inclusion of the meteorological variables. Again, the cluster of SIF, LAI, and FPAR shows by far the highest importance, followed by the PAR, RSDN, ET, and temperature-related variables (Fig. 7). The meteorological variables temperature, VPD, and precipitation are generally in clusters of lower importance, as are the MODIS NBAR features. In contrast, the "RS minimal" product shows the highest variable importance for the visible NBAR spectrum, followed by NIR and PAR in descending order. The SWIR bands are hardly used in any setup.

Furthermore, we grouped the predictions by site and evaluated the site-level r² for each land cover type for AutoSklearn with "RS" explanatory variables (Fig. 8). EBF and SH sites show low r² (median r² -0.38 and 0.33, respectively) with substantially higher variance, whereas MF and DBF could be predicted with high quality (median r² 0.84 and 0.87, respectively). Regarding anomaly estimation, EBF and WET show significantly lower r² values (median r² 0.04 and 0.01, respectively). Furthermore, our analysis indicated that models tended to exhibit a significant positive bias when predicting small GPP values (in the lowest quartile) while displaying a negative bias for large GPP values. This implies an overestimation of small GPP and an underestimation of large GPP values by the models.

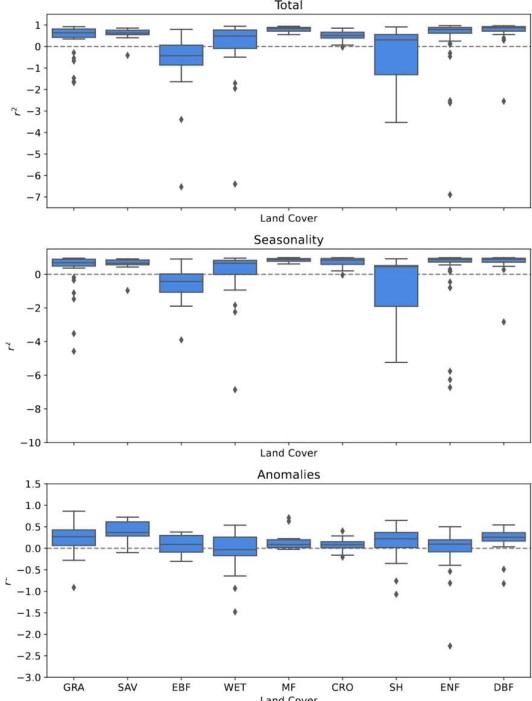

**Figure 8 Distribution of r² values for the GPP prediction by AutoSklearn with "RS" explanatory variables for different land cover types. Shown are the overall performance and performances for seasonality and anomalies.**

Finally, we examined the effect of including higher-resolution data in the explanatory data. Replacing the MODIS reflectance bands, LAI, FPAR, and land cover products with their 500 m resolution counterparts resulted in significant improvements in r². We tested this behavior for AutoSklearn with the "RS" variable set. The prediction r² was with $0.8164 \pm 0.0005$ overall and $0.444 \pm 0.003$, $0.787 \pm 0.002$, $0.8723 \pm 0.0005$, and $0.3094 \pm 0.0006$ for trend, across-site variability, seasonality, and anomalies, respectively, in all aspects except trend significantly higher than for the lower resolution data product (Fig. 9).

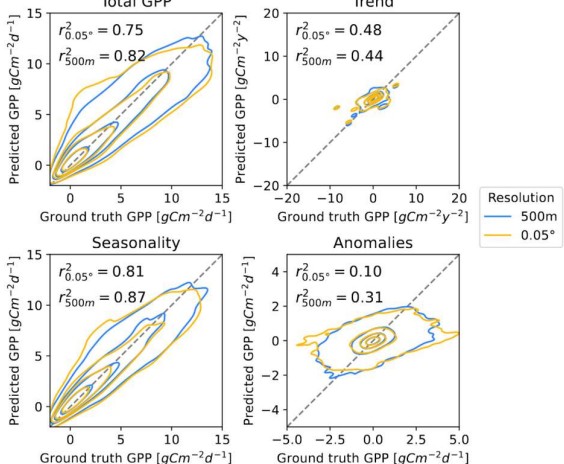

**Figure 9 Comparison of the predicted 0.05 ° product and the one with 500 m resolution from AutoSklearn ensemble averages and the "RS" variable set. The latter shows higher r² values compared to the ground truth GPP estimates from FLUXNET, AmeriFlux OneFlux, and ICOS. We refer to GPP measurements derived from eddy covariance at the flux tower locations as ground truth.**

### 3.2 Analysis of AutoSklearn Pipelines

We investigated the different components (base models and preprocessing algorithms) of the AutoSklearn framework, which was trained on the "RS" variable set in the repeated cross-validation (See figure A4 for the model run statistic). For every fold in each of the repeated cross-validations, we considered the best-performing model of each base-model type and min-max-scaled their RMSE to a scale from zero to one. The scaling accounts for the different predictability of the test data in the

respective fold. We then took the mean across all folds within each repetition of the cross-validation and each base-model type, resulting in a distribution of scaled RMSEs for each base-model type (Fig. 10). We also considered whether these models preprocessed the training data or not.

The base models achieving the lowest scaled RMSE were ensembles of weak learners, such as Extra Trees, Random Forest,
Gradient Boosting, or AdaBoost. These models could, by themselves, achieve the best predictions of GPP. That, however, does not suggest that they were necessarily used in the final model ensemble constructed by AutoSklearn. The ensemble selection algorithm (forward stepwise model selection) in AutoSklearn, which creates the model ensembles, recursively adds the base models that improve the RMSE of the ensemble prediction most in combination with the models already part of the ensemble (Caruana et al., 2004). Hence, a model showing a low RMSE by itself does not need to be beneficial to the ensemble
of models ultimately used by AutoSklearn.

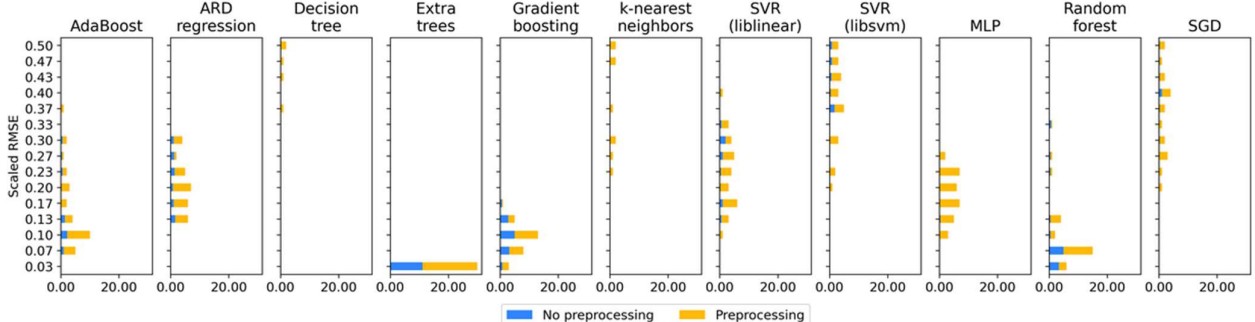

**Figure 10 Performance of AutoSklearn base models and feature pre-processors. The chart shows the distribution of the mean RMSE for each base model type across all folds within each repetition of the cross-validation. We considered only the best-performing models for each model class within each fold. The RMSE is min-max scaled from zero to one within each cross-validation fold to**
**account for variations in the data's predictability depending on the data's split. The use of preprocessing algorithms is shown as colors in the proportions of their usage in each bin (detailed preprocessing methods in figure A5).**

### 3.3 Global GPP maps

From the bootstrap aggregation of the AutoSklearn framework with "RS" features, we predicted global GPP with wall-to-wall coverage, resulting in 30 predictions for the entire period from 2001 to 2020 in monthly intervals. In addition, we applied land-
sea and vegetation masks to the prediction, similar to previous research (Tramontana et al., 2016; Joiner et al., 2018).

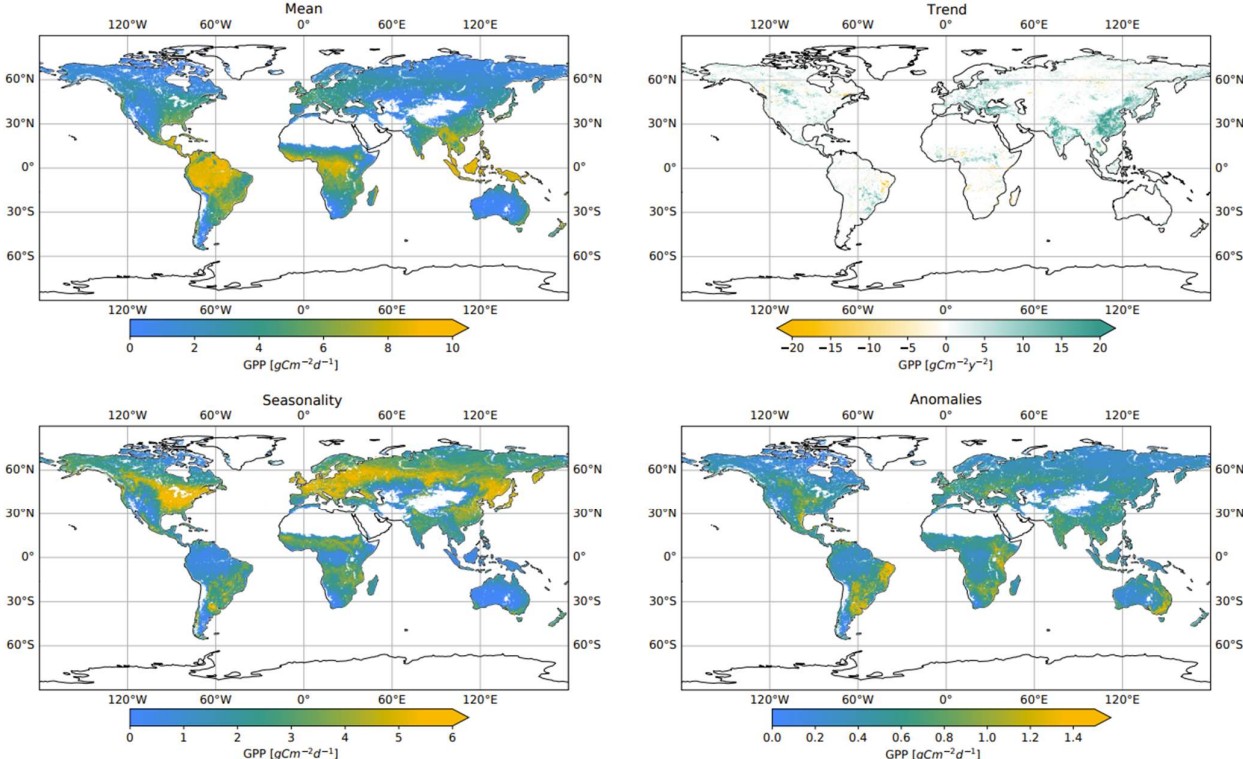

**Figure 11 Total GPP, amplitude of seasonality, trend, and anomalies of prediction with AutoSklearn trained on remotely sensed data ("RS" dataset) in a bootstrap aggregation of 30 bootstraps. The mean was calculated at each location over all bootstrapped predictions and the entire time series. The seasonality is displayed as the amplitude of the month-wise average. Trends were calculated as the slope from an ordinary least squares linear regression over time and masked so that only significant trends were included ($p < 0.05$). The anomalies are shown as the standard deviation of the residuals after subtracting the seasonal and trend components from the time series.**

Mean GPP for 2001–2020 (Fig. 11) showed high values for tropical climates in low latitudes, such as the Amazon region, Southeast Asia, and Central Africa, with maximum GPP values for the EBF land cover. Conversely, low GPP appears in high latitudes and SH, SAV, and GRA regions.

Again, we decomposed the local time series into trends, seasonality, and anomalies (Fig. 11). The amplitude of the seasonal component exhibits significant regional differences. Mid-latitude regions in the northern hemisphere show high amplitudes, covering the central and eastern US, Europe, parts of Russia, and north-eastern China. In contrast, low-latitude regions have low GPP amplitudes. The data show significant trends ($p < 0.05$) over the observation period with positive clusters, especially for eastern China and western India, while negative trends are less pronounced. The bootstrapped AutoSklearn framework shows clusters of high GPP anomalies in, e.g., parts of South America (especially eastern Brazil and Argentina), East Africa, and Southeast Australia. Land cover in these areas does not follow a consistent pattern but is often dominated by CRO, SH, and GRA.

In addition to the GPP prediction, we produced a sampling error estimate by calculating the average standard error across all bootstraps for each location and time (Fig. 12). We observed high relative errors in low GPP regions, high latitude regions (e.g., with temporary snow cover), and arid SH regions. The distribution of standard errors relative to the bootstrap mean peaks near zero and ends in a long tail towards higher values for all biomes (Fig. 13a). However, the distribution of sampling uncertainty in GPP varies among land cover classes, ranging from low medians for EBF (0.5 %) and SAV (0.8 %) up to higher medians for ENF (4.0 %) and SH (6.9 %).

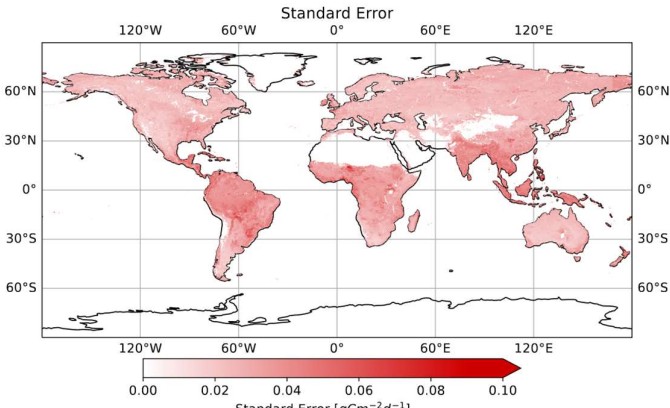

**Figure 12 Absolute standard errors from the bootstrap aggregation. For relative values, see figure A6.**

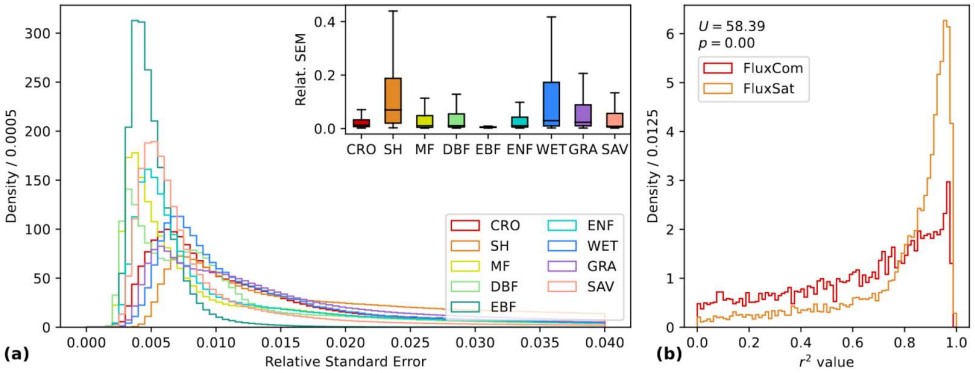

**Figure 13 Histogram of the relative standard error of the mean (SEM) by land cover class during the entire observation period (a) and distribution of $r^2$ values for total GPP of the upscaled GPP AutoSklearn product with "RS" variables, compared to the FluxCom v6 and FluxSat datasets (b). For the latter, GPP is sampled at 10000 random locations and compared in a Mann-Whitney U test.**

### 3.4 Comparison to reference data

We compared the upscaled results of total GPP from our AutoSklearn "RS" prediction with GPP datasets FluxCom v6
(Tramontana et al., 2016) and FluxSat (Joiner et al., 2018) at 10000 random sample locations. When tested with a Mann-Whitney U test, our predictions show significantly higher agreement (p virtually zero) with FluxSat than with FluxCom (Fig. 13b). In our prediction, 51 % of the samples explain more than 80 %, respectively, of the variation in FluxSat, while this is the case for only 17 % of the samples in FluxCom. Thus, AutoSklearn shows good agreement with the GPP patterns predicted by FluxSat, whereas it deviates more strongly from the FluxCom product.

### 4 Discussion

### 4.1 AutoML framework performance

The results demonstrate the closeness of the overall predictive performance of the evaluated frameworks and the baseline Random Forest. Despite the different complexity of the model architectures, the frameworks capture a similar fraction of the variability in the GPP measurements. Framework choice does not appear to be a major factor in this experimental setup,
resulting in only a low difference in $r^2$. These findings align with previous research on applying classical ML models (Tramontana et al., 2016).

The performance differences between the frameworks are statistically significant but slight. AutoSklearn consistently outperforms H2O AutoML, AutoGluon, and Random Forest. The framework is based on ensemble prediction, which can
exploit the different advantages of each base model. The evaluation of base models used by AutoSklearn outlines the

applicability of various ML model types for predicting GPP. It is evident that ensembles of weak learners, such as Extra Trees or Random Forest, are generally favorable for this task. These models can be promising for GPP prediction either in a stand-alone implementation or as part of a model ensemble. The performance comparable to H2O AutoML and AutoGluon shows furthermore that implementing feed-forward neural networks does not necessarily lead to performance improvements. Low performance of AutoGluon, even when compared to Random Forest, may relate to the lack of hyperparameter tuning. However, the differences between frameworks are challenging to explain, as the reasons for the frameworks' results are obscured by their black box character.

AutoSklearn trained on "RS" explanatory variables tended to overestimate small GPP values while underestimating large GPP values. This behavior was already observed in the FluxCom (RS), FluxSat, and several light use efficiency models (Yuan et al., 2014; Joiner et al., 2018). It has also been shown for the early MODIS GPP product (Running et al., 2004), where the overestimation was attributed to an artificially high FPAR while the underestimation was related to low light use efficiency in the MODIS algorithm (Turner et al., 2006). Another reason could be the strong reliance of the AutoSklearn framework on tree-based models (Fig. 10). These models are constructed by recursively partitioning the feature space into small regions to which they fit a simple model, which limits them in their ability to extrapolate beyond the range of target values already observed. Furthermore, our predictions showed differing prediction quality at the land cover level, which might result from biome-specific circumstances and the availability of measurement sites. For example, biomes with a pronounced seasonal cycle, such as DBF or MF, exhibit high overall $r^2$, whereas EBF and WET show large variability that the model could not capture. In addition, variability within a land cover type could affect the performance assessment, such as for SH, which includes both arid and subarctic shrublands.

Finally, it is crucial to note that the $r^2$ metric only expresses how well a framework can reproduce measurements from the measurement samples, which are limited in underrepresented areas. We grouped data by site and applied a land cover stratification during the CV to increase independence between the folds. That, however, does not prevent sites from being repeatedly selected for validation during the repeated CV, which can inflate the performance metric and reduce variance. It also cannot account for spatial autocorrelation. This affects the assumption of independence and identical distribution for train and test folds, which is crucial for obtaining realistic CV results. Violating these requirements can lead to overestimating model performance and inflating map accuracies, yet it is commonly done in data upscaling efforts (Roberts et al., 2017; Ploton et al., 2020). More training data with better geographic representation could help mitigate these shortcomings and could lead to more robust predictions, model evaluations, and potentially higher model performance.

### 4.2 Importance of explanatory variables

AutoML is a powerful approach for assessing the importance of the variables on model performance since it selects the optimal base models and constructs optimal pipelines independently for each feature set under consideration. This means that no subjectivity bias is introduced into assessing variable importance, e.g., by pre-selecting specific algorithms that are expected to perform well on a particular task or set of explanatory variables. This could increase the quality of the reported importance, especially as features in GPP prediction often exhibit severe intercorrelations. Importantly, variable importance is model-specific, meaning it can indicate which variable is most effectively used by a particular model, but it does not directly indicate the intrinsic predictive value of a variable. Furthermore, it may depend on the choice of temporal and spatial scales and data quality, given that many of the input features are themselves model outputs.

The frameworks' performance depends significantly on the choice of predictive features on which they are trained. The results show that while the seven NBAR bands and PAR from the "RS minimal" variable set provide the model with sufficient

information for a GPP prediction, the full set of "RS" variables adds additional information that all the frameworks can exploit. The additional variables in the "RS" variable set, such as SIF, LAI, FPAR, ET, LST, SM, and plant function type, appear to include important environmental forcings and structural variables that provide a marginal advantage over the variables on only vegetation structure and radiation in "RS minimal" (Green et al., 2019; Stocker et al., 2019; Xu et al., 2020). For example, environmental stress, such as heat waves and droughts, often causes instantaneous reductions in GPP. However, the response of vegetation greenness to these stressors is typically slower and may only become apparent if the stress persists for a sufficient duration (Orth et al., 2020; Zhang et al., 2016; Smith et al., 2018; Yan et al., 2019). In such cases, relying solely on surface reflectance may not sufficiently capture the variability of GPP.

Including the meteorological explanatory features (ERA5-Land) in the training data does not significantly improve the prediction quality for any of the frameworks. This implies that meteorological data may not contain additional information that the machine learning frameworks in this study can effectively use to predict GPP. A possible explanation could be the mismatch between reanalysis and site meteorology. The coarse resolution and large uncertainties of the reanalysis data may result in a poor representation of the flux tower footprints, which are often smaller than one pixel of the reanalysis data, leading to uncertainties in the modeling. For example, Joiner and Yoshida (2020) showed that using site-measured meteorological data instead of reanalyzed data significantly improved the performance of GPP predictions. At the monthly scale, the "RS" variable set may already encode information about the instantaneous environmental stress from adverse meteorological conditions through, for example, LST, ET, and soil moisture, which are important controls on GPP (Bloomfield et al., 2023). Further studies could potentially assess these uncertainties by comparing models trained with tower meteorological data to gridded reanalysis datasets.

The permutation importance of explanatory variables provides further insight into which variables AutoSklearn uses and which are indifferent to the framework. Our results show that both "RS" and "RS meteo"-trained AutoSklearn frameworks rely primarily on features of canopy structure (LAI, FPAR), proxies for photosynthetic activity (SIF), and ET, which strongly couples with GPP in favorable environmental conditions. Meteorological information, such as temperature and VPD, are less relevant for the model prediction. This suggests that the insignificant changes in performance between "RS" and "RS meteo" may be related to a small additional contribution of meteorological conditions to the prediction of monthly GPP beyond what is already provided by vegetation structure and PAR. Soil moisture was also found to have minimal influence overall, which might be partly due to uncertainties and noises in the remote sensing soil moisture data and due to its coarse spatial resolution. It is also important to note that previous studies have demonstrated the importance of soil moisture from SMAP in predicting GPP in water-limited ecosystems (Dannenberg et al., 2023; Kannenberg et al., 2024). The performance difference between "RS minimal" (NBAR and PAR only) and "RS" variables seems to be driven at least partly by features that are themselves model outputs based on MODIS NBAR, i.e., SIF, LAI, and FPAR. We grouped the variables into clusters with high correlation to improve the interpretability of the importance measures. However, we could not completely eliminate correlations between clusters. High correlations between, for example, PAR and LST, and ET and PAR, as well as lower correlations between other variables, could not be taken into account and introduced further uncertainty in the reported variable importance.

The ability of the frameworks to reproduce GPP patterns and the corresponding variable importance must be evaluated in light of the choice of temporal resolution. In this study, we evaluated machine learning upscaling of monthly GPP dynamics, which are dominated by light availabilities and seasonal changes in vegetation structures. However, at shorter time scales, such as hourly or daily, GPP is more closely aligned with diurnal and short-term variations in meteorological conditions such as temperature and VPD. Thus, these variables are likely more influential in predicting GPP at these higher frequencies (Frank et al., 2015; von Buttlar et al., 2018). Additionally, complex machine learning models may also offer greater benefits at

harnessing the large data quantities involved in predicting GPP at hourly or daily scales. Further research is needed to benchmark machine learning algorithms and assess choices of environmental data in predicting GPP across different timescales.

560 We found that besides selecting an appropriate set of explanatory variables, the resolution of the data highly affects prediction outcomes. Including 500 m resolution data should reduce the mixed pixel problem and match the flux towers' footprints better with the pixel size of the gridded data sets. This led to improvements in all time series components, with exceptional increases in $r^2$ for the estimation of anomalies. These results underscore the importance of spatial resolution and suggest the use of data with a resolution that better represents smaller landscape features and flux tower footprints, in contrast to our initial choice of 565 0.05 ° resolution in this study.(Xiao et al., 2008; Yu et al., 2018; Chu et al., 2021).

### 4.3 Spatio-temporal patterns

The globally upscaled measurements could capture the variation of GPP in the ML-based FluxCom and FluxSat reference datasets reasonably well and resemble their total GPP patterns and seasonality (Tramontana et al., 2016; Joiner and Yoshida, 2021). However, the prediction could explain a significantly larger fraction of the variation in FluxSat than in FluxCom. Both 570 datasets are based on MODIS-derived products, but the training sites we used show higher similarities to FluxSat than to FluxCom.

We observed several clusters of positive trends consistent with previous results and local studies (Chen et al., 2019; Wang et al., 2020; Schucknecht et al., 2013; Carvalho et al., 2020). However, the magnitude was lower than the reference dataset 575 FluxSat (Joiner and Yoshida, 2021) and showed less frequent significant negative trends than predicted by FluxCom (Tramontana et al., 2016). The areas with high predicted GPP overlap with the highly productive regions in the tropics and mainly cover the EBF regions (Ahlström et al., 2015). In addition, we observed high seasonality, especially in CRO-dominated regions, which may be due to high productivity in maize, wheat, rice, and soybean cultivation and a profound seasonality, with a period of very low GPP after harvest. (Kalfas et al., 2011; Gray et al., 2014; Sun et al., 2021). High anomalies occurred in 580 mainly temperate and semi-arid climates, the latter of which have also been shown to dominate the interannual variability of the global terrestrial carbon sink (Ahlström et al., 2015). Besides random variations included in the anomalies, reasons could be non-seasonal events, such as weather extremes or human interventions, coupled with a high turnover rate in dry vegetation. The patterns agree with FluxSat and exceed those that FluxCom models estimated.

### 4.4 Uncertainty

Predicting wall-to-wall maps from a non-representative distribution of measurement sites is challenging. A non-representative network of flux towers might fail to reproduce the main features of the underlying GPP population for the entire study area (Sulkava et al., 2011). Land cover types with less abundant eddy covariance measurements may potentially be estimated less reliably and could show a higher variation in GPP estimations. We used the standard error to estimate how robustly the frameworks react to different subsets (bootstraps) of data during the training process. Generally, high relative error values in 590 low GPP regions are expected due to the normalization of the error. However, SH, ENF, and regions adjacent to SNO and BAR also show an elevated error in absolute terms. The distributions (Fig. 13a) show similarities to the spread of $r^2$ values obtained from the framework benchmark (Fig. 8).

Higher standard errors may indicate that monthly remote sensing and modeled input data are better proxies for some 595 ecosystems than others. For example, GPP can be predicted with low relative uncertainty for ecosystems with a high seasonal variation of biomass, such as croplands, broadleaf forests, and mixed forests. In contrast, predicting GPP in drylands can be

more challenging. Drylands are highly sensitive to water availability, resulting in abrupt responses to precipitation and drought events (Barnes et al., 2021). They are characterized by high spatial heterogeneity and irregular temporal vegetation patterns, which are difficult to capture at our spatial and temporal resolution. Together with a low vegetation signal-to-noise ratio, these factors pose a considerable challenge for GPP remote sensing (Smith et al., 2019). In an attempt to assess the uniqueness of NEE measurements at FLUXNET sites, Haughton et al. (2018) showed that drier sites and shrubland sites had a higher discrepancy between locally and globally fit models and exhibited more idiosyncratic NEE patterns compared to others. Our results show a similar behavior, with higher model uncertainty for GPP in dryland and shrubland regions.

The results delineate that AutoSklearn could not reliably infer a robust functional relationship in low-productivity regions, where it shows a significant positive bias. We suggest further research on ways to improve performance in low-GPP regions. One method that could potentially enhance the prediction is to include dummy measurement sites in the masked regions manually. These sites would constantly report zero GPP and could improve estimates in adjacent regions, such as arid zones or seasonally snow-covered areas, which are also less proportionately represented in the flux tower networks (Smith et al., 2019).

Finally, an additional limitation is introduced by the eddy covariance measurements themselves. We use night-time-partitioned GPP, which is modeled as the difference between NEE and ecosystem respiration. While NEE and night-time respiration are directly measurable, daytime respiration is modeled with a temperature response function, which extrapolates from night-time respiration (Reichstein et al., 2005). Up to this point, it is not conclusively clarified how reliably this approach can be employed, considering that it is indifferent to some environmental stress factors and changes in respiration behavior between day and nighttime (Wohlfahrt and Galvagno, 2017; Keenan et al., 2019; Tramontana et al., 2020). The inherent uncertainty and bias in the ground truth GPP data could be a potential cap to the performance we can obtain in our efforts to predict GPP.

**5 Conclusion**

We investigated whether and how automated machine learning (AutoML) frameworks can improve global upscaling of gross primary productivity (GPP) from *in situ* measurements using AutoSklearn, H2O AutoML, AutoGluon, and a baseline Random Forest model in repeated cross-validation stratified by land cover. In addition, we evaluated different sets of explanatory variables for the GPP prediction from satellite imagery and ERA5-Land reanalysis data. Our results show that the AutoML frameworks can capture about 70–75 % of the monthly GPP variability at the measurement sites.

AutoSklearn slightly but significantly outperformed the other frameworks across all sets of explanatory variables for total GPP, trends, seasonality, and anomalies. It did this by creating ensembles of base models and preprocessing algorithms that improved the prediction over individual machine learning models. The ensemble members were primarily models that combined weak learners, such as Extra Trees, AdaBoost, or Random Forests. However, the difference in performance was small compared to other frameworks and the Random Forest model, suggesting that the choice of framework may play only a minor role in improving GPP prediction performance.

We found that remotely sensed (RS) explanatory variables provided the best results in combination with the investigated frameworks. While only relying on the MODIS NBAR reflectance bands and PAR ("RS minimal") provided the models with sufficient information for GPP prediction, considering other proxies of photosynthetic activity and canopy structure, such as solar-induced fluorescence, leaf area index, and fraction of absorbed photosynthetic activity, increased the performance of all models. Meteorological factors and soil water availability had less influence on the GPP prediction. Also, additional

meteorological variables from ERA5-Land could not be used effectively by the models. In particular, the resolution of the satellite imagery played a significant role in prediction quality.


Finally, we used the best-performing framework (AutoSklearn with "RS" explanatory variables) to upscale GPP to global wall-to-wall maps in a bootstrapping approach. The predictions are in good agreement with the FluxSat dataset and deviate significantly more from the FluxCom predictions. The GPP product captures major spatial patterns for total GPP and trends but shows high uncertainty for low-GPP regions, where the predictions are positively biased. In general, prediction

performance and sampling uncertainty are highly dependent on the land cover type.

In conclusion, AutoML can be a considerable technique for predicting and extrapolating GPP from *in situ* measurements. Automated creation of machine learning pipelines can facilitate the process of algorithm and feature selection, thereby avoiding biases in the modeling process. In addition, AutoML enables the exploration of a wide range of models and algorithms,

uncovering potential relationships and patterns that may have been missed manually. However, we were unable to demonstrate that AutoML produces GPP predictions that are considerably more accurate and robust than classical ML models. In particular, the non-automated Random Forest model performed almost as well as AutoSklean. Researchers must carefully interpret and validate the results obtained through AutoML, ensuring that the models and features chosen are consistent with ecological knowledge and scientific understanding. Nevertheless, given the early stage of development, AutoML may be useful in the

future to improve and accelerate research on GPP upscaling.

**Appendix**

**Equation A1 Coefficient of determination $r^2$, where $y_i$ is the observed value, $\hat{y}_i$ the modeled value, and $\bar{y}$ the observed average over all $N$ values.**

$$r^2 = 1 - \frac{\sum_{i=1}^{N}(y_i - \hat{y}_i)^2}{\sum_{i=1}^{N}(y_i - \bar{y})^2}$$

**Table A1 Overall framework performance. Shown are the mean $r^2$ values with the corresponding error of the mean, averaged over all cross-validation repetitions. Additionally to the three predictor variable sets, we added the vegetation indices (VI) NDVI (Normalized difference vegetation index), EVI (Enhanced vegetation index), GCI (Green chlorophyll index), NDWI (Normalized difference water index), NIRv (Near-infrared reflectance of vegetation), and kNDVI (Kernel NDVI) to each variable set to evaluate if they improve the performance.**

| Variable set | Random Forest | H2O AutoML | AutoSklearn | AutoGluon |
|---|---|---|---|---|
| RS minimal | 0.7052 ± 0.0003 | 0.7112 ± 0.0009 | 0.7214 ± 0.0005 | 0.7013 ± 0.0005 |
| RS minimal (incl. VI) | 0.7193 ± 0.0002 | 0.7166 ± 0.0007 | 0.7261 ± 0.0004 | 0.7097 ± 0.0007 |
| RS | 0.7369 ± 0.0002 | 0.739 ± 0.001 | 0.7452 ± 0.0003 | 0.7324 ± 0.0003 |
| RS (incl. VI) | 0.7352 ± 0.0002 | 0.7383 ± 0.0004 | 0.7437 ± 0.0003 | 0.7315 ± 0.0002 |
| RS meteo | 0.7383 ± 0.0002 | 0.7416 ± 0.0008 | 0.7214 ± 0.0004 | 0.7318 ± 0.0004 |
| RS meteo (incl. VI) | 0.7356 ± 0.0002 | 0.7402 ± 0.0005 | 0.7201 ± 0.0003 | 0.7310 ± 0.0002 |


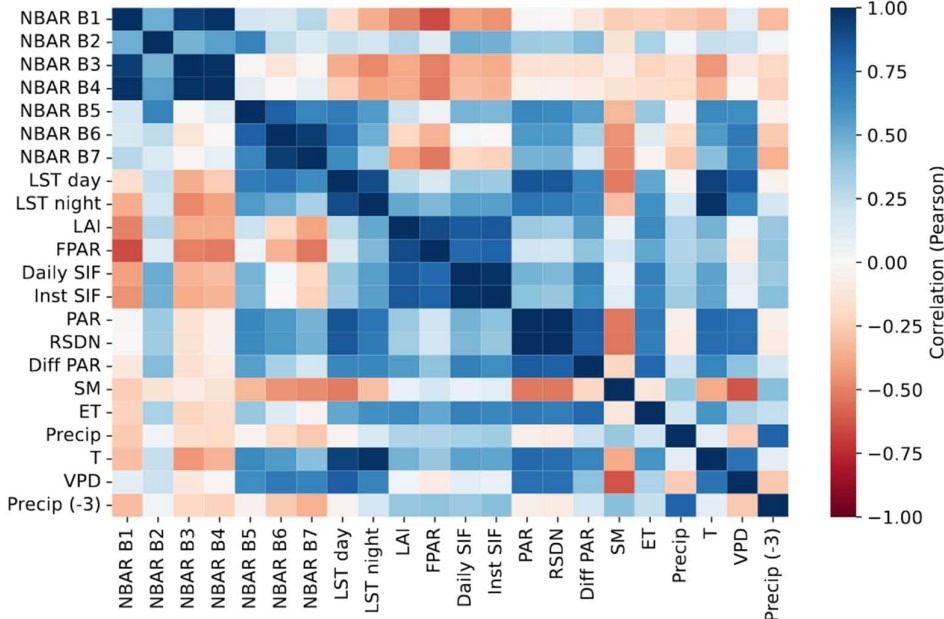

**Figure A1 Pearson correlation matrix between the scalar explanatory variables, including the MODIS NBAR bands, land surface temperature (LST), leaf area index (LAI), photosynthetically active radiation (PAR), fraction of absorbed PAR (FPAR), diffuse PAR (Diff PAR), daily and instantaneous solar-induced fluorescence (SIF), surface downwelling shortwave flux (RSDN), soil moisture (SM), evapotranspiration (ET), precipitation (Precip), temperature at 2m height (T), vapor pressure deficit (VPD), and precipitation with 3-months lag (Precip (-3)).**

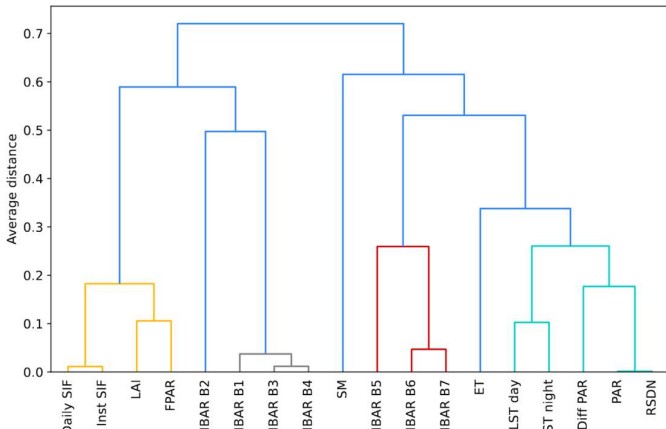

**Figure A2 Dendrogram for clustering the explanatory variables of the "RS" set. The variables are clustered after their average distance, which is one minus the absolute of the Pearson correlation coefficient. See figure A1 for variable abbreviations.**

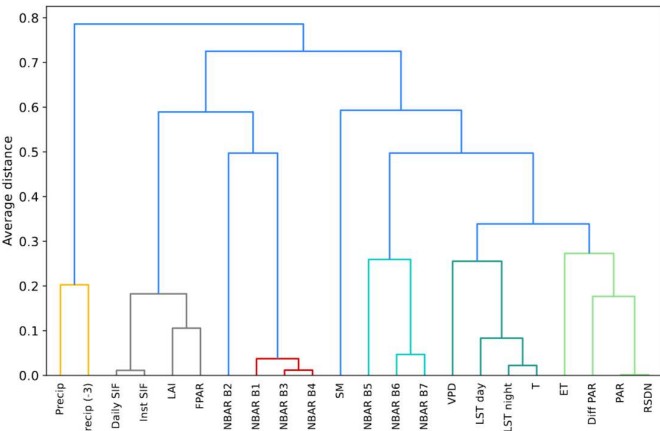

**Figure A3 Dendrogram for clustering the explanatory variables of the "RS meteo" set. The variables are clustered after their average distance, which is one minus the absolute of the Pearson correlation coefficient. See figure A1 for variable abbreviations.**

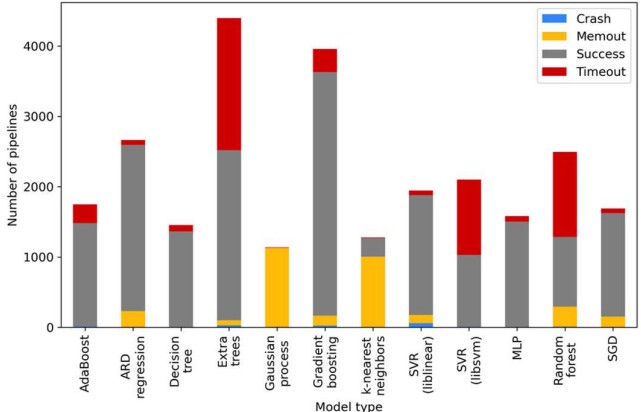

**Figure A4 Run statistics of the AutoSklearn base models. The four statuses show how many base models succeeded or failed during training due to insufficient memory, training time, or other unknown reasons. Only the successful models were used for the configuration of AutoSklearn.**

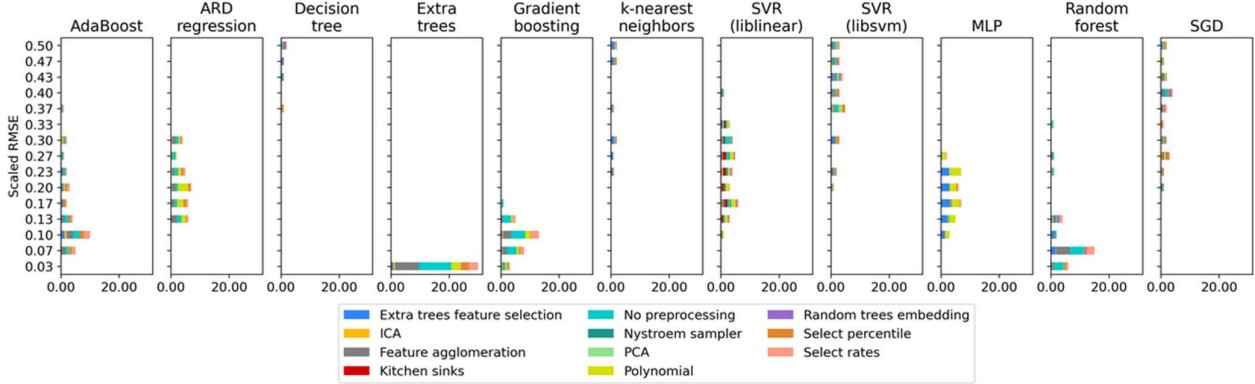

**Figure A5 Detailed use of preprocessing algorithms by AutoSklearn. The chart shows the distribution of the mean RMSE for each**
**base model type across all folds within each repetition of the cross-validation. We considered only the best-performing models for each model class within each fold. The RMSE is min-max scaled from zero to one within each cross-validation fold to account for variations in the data's predictability depending on the data's split. The use of preprocessing algorithms is shown as colors in the proportions of their usage in each bin.**

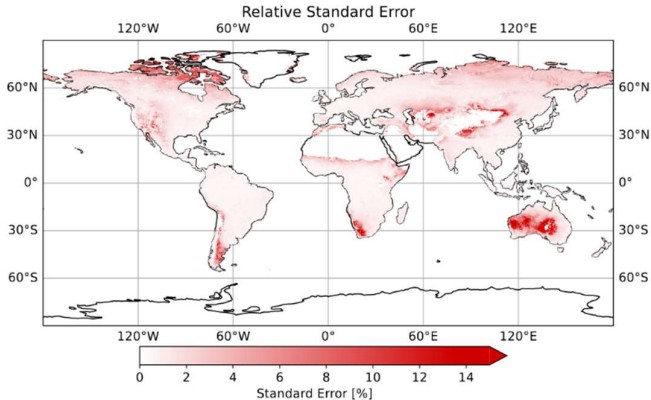


**Figure A6 Relative average standard error, normalized by the mean GPP prediction.**

**Code availability**

The code can be found at 10.5281/zenodo.8262618.

**Author contribution**

The study was conceptualized by YK and MG. YK contributed to the data curation. MG performed the formal analysis and developed the experimental methodology. MG prepared the manuscript draft, with contributions from YK and the other co-authors. The project was supervised by TK and GS.

**Competing interests**

The authors declare that they have no conflict of interest.

**Acknowledgments**

We would like to express our gratitude to Martha Anderson and Christopher Hain for providing the ALEXI ET dataset, which has greatly enriched our research. TK acknowledges funding from the LEMONTREE (Land Ecosystem Models based On New Theory, obseRvations and ExperimEnts) project, funded through the generosity of Eric and Wendy Schmidt by recommendation of the Schmidt Futures programme, a DOE Early Career Research Program award #DE-SC0021023, and 705 NASA Awards 80NSSC21K1705 and 80NSSC20K1801. YK acknowledges support from a DOE Early Career Research Program award #DE-SC0021023 and the LEMONTREE project.

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
