# Peer review of "Using automated machine learning for the upscaling of gross primary productivity"

_Biogeosciences, 2023_

## Author Comment (AC1)

**Responses to referee #1**

*Max Gaber and colleagues investigate the effect of several technical choices in the process of predicting GPP from eddy-covariance measurements and satellite(-derived) data sets using machine learning. The focus is on novel methods in the field of automated machine learning applied to predict monthly GPP at site level from different sets of predictor variables, as well as on the effect of their spatial resolution. The authors demonstrate the applicability of AutoML, and of AutoSklearn in particular, and show that in the global upscaled product, spatiotemporal patterns reasonably compare to other products. They also illustrate the importance of adequate spatial resolution of the predictor variables by increased model performance at site level, when part of the predictor variables are fed into the machine learning at 500m instead of at 0.05deg resolution.*

*Given the growing number of research studies that implement such data-driven approaches at global and regional scales (large part of whom are cited in the paper) and the still unquantified importance of several technical choices in the set-up, this study is timely and definitely of relevance. The fact that the overall R2 at site level is similar to or slightly higher than from a plane random forest or the results in comparable upscaling exercises at monthly scale (Jung et al. 2011) is interesting, and highlights that tuning the machine-learning set-up may not be the most promising way forward to improving the performance of data-driven models, but rather more informative predictor variables (at least Fig.5 may be interpreted in this way) . I find this a very valuable finding which may also deserve to be communicated/ highlighted more clearly (like you did for example in l. 384-389, but not in the abstract or elsewhere). At the moment the differences in significance are stressed more than the very similar magnitude of performance between the different AutoML methods. Also, your finding that the AutoML does not help to reproduce interannual changes (l. 267) is an important finding, because it is a common problem in data-driven upscaling and very relevant question in the carbon cycle community, and therefore in my opinion deserves to be stressed more.*

Response: Thank you for your valuable feedback! We appreciate your constructive comments, which are very helpful in improving the manuscript. Indeed, tuning the machine learning setup might only produce marginal improvements compared to considering more informative predictor variables. We will highlight this finding more prominently in the text. We will, furthermore, stress the problem of AutoML to reproduce interannual variabilities, as suggested.

*I suggest publication of the paper after addressing the following major questions/ comments:*

*1. What is the reason for doing this analysis at a monthly temporal scale when structural vegetation changes dominate rather than finer temporal resolution? I would expect higher gains from AutoML and also more differentiated contributions between predictor variables (especially meteorological features) at higher temporal resolution. This is also the time scale which is more relevant to be able to properly represent seasonal and anomalous trajectories. I would expect large potential from automated model tuning especially for short extreme events, which are relevant for the carbon uptake and hard to represent in a data-driven model set-up, but clearly smeared out at a monthly time step. Much of the discussion in section 4.2 does neglect the coarse time step when for example LUE changes are not expected to play major role.*

Response: The temporal resolution is indeed an important factor in the contribution of the different predictor variables. A higher temporal resolution could enable the models to represent better anomalies, extreme events, and their impact on GPP (see, f.x. Bodesheim et al. (2018)). Since many previous upscaling works focus on monthly scales, and these data have been instrumental in

informing global long-term dynamics of GPP across different regions in many studies, we have chosen to perform this evaluation at monthly scales as an initial step. Our team has follow-up studies that examined more advanced machine learning algorithms, such as the temporal fusion transformer (TFT), in modeling the dynamics of GPP at hourly scales across space (Rumi Nakagawa et al., 2023). More assessments are necessary to quantify machine learning performance under different time scales. We will make sure to highlight better the consideration of temporal scales on the upscaling framework and model choice in our discussion in the revised manuscript.

References:

Bodesheim, P., Jung, M., Gans, F., Mahecha, M. D., and Reichstein, M.: Upscaled diurnal cycles of land–atmosphere fluxes: a new global half-hourly data product, Earth System Science Data, 10, 1327–1365, https://doi.org/10.5194/essd-10-1327-2018, 2018.

Nakagawa, R., Chau, M., Calzaretta, J., Keenan, T., Vahabi, P., Todeschini, A., Bassiouni, M., Kang, Y., 2023. Upscaling Global Hourly GPP with Temporal Fusion Transformer (TFT). https://doi.org/10.48550/arXiv.2306.13815

***2. Data sets:***

***A number of predictor variables are model outputs themselves, relying on input data and model assumptions. This is not discussed at all.***

Response: We will include further discussion about the sources of the variable input with a focus on introduced uncertainty from the modeling process.

***What is the reason for ingesting both SIF and instantaneous SIF, or both PAR and RSDN?***

Response: Our approach was to include as many predictor variables as possible and let the AutoML frameworks decide themselves what variables are necessary for a good prediction. This includes variables showing a high intercorrelation and potentially small differences in predictive capacity. We will provide clarification in the revised manuscript.

***How is the temporal aggregation done?***

Response: We aggregated with a simple average within the respective period after filling the data gaps (see below). We will clarify this in the text.

***How do you handle data gaps?***

Response: We filled gaps at native temporal resolution. For high-resolution data products (frequency <=4 days), such as NBAR, LAI/FPAR, BESS, CSIF, and CCI, we filled gaps less or equal to 5 days (8 days for products with a 4 day resolution) with the average of a 15-day moving window. We gap-filled LST with a 9-day moving window since we observed higher variations. Soil moisture was filled after Walther et al. (2021) with moving window medians for short gaps and mean seasonal cycle for long gaps. We will clarify this in the text.

***Handling of bad data quality is only mentioned for the site-level fluxes, what about the explanatory variables?***

Response: We used NBAR, where >75% high-resolution NBAR pixels were available from full BRDF inversion. We applied the quality control mask for LST where the average emissivity error is < 0.02. LAI/FPAR was used with and without saturation. We used all data for soil moisture. We will include this in the text.

*Specify more clearly the data sources, e.g. for the CCI soil moisture, which version did you use? Presumably, FluxCom v6 refers to the FluxCom set up with RSonly ( only satellite-based predictors using MODIS collection 6), which is 8-daily and at high spatial resolution?*

Response: We used CCI Soil moisture v.06.1 and FluxCom v6 RS only. We will include this in the text.

*3. Spatial resolution: Why not also ingest tower meteorology instead of the coarser ERA5-Land? The scale mismatch could be further discussed, especially between a 0.05deg pixel and the tower footprint. The way the authors approach the analysis suggests using the 0.05deg pixel is the generally accepted default, which is not the case.*

Response: Thank you for raising this interesting point. The spatial mismatch is a large uncertainty factor in the prediction, as outlined in the manuscript (l.309-314). Tower meteorology is expected to increase predictive performance substantially compared to the coarse-resolution ERA-5 product. Regarding using meteorological variables as predictor variables for global upscaling, however, tower meteorology poses a limitation due to its spatially constrained availability. It cannot be used as a predictor for regions where no flux tower data exists. For this reason, we chose ERA-5 land, since it is globally available and, hence, can be used for global predictions. It would be interesting to evaluate uncertainties in reanalysis data using tower meteorology and understand the potential impacts on upscaling uncertainties. We will clarify and discuss this aspect in the text.

*4. In parts the manuscript uses very technical language and describes key concepts only in a very short manner. I suggest to rephrase certain passages to make the manuscript better accessible to a wider audience which may also not be very familiar with the newest developments in the machine learning world – or at least expand more in the supporting information. Examples of very technical sentences in my opinion are l.160-161, l.165-166, l.170-172, l.177-181, l.243-246*

Response: We will make the text more accessible and reformulate the mentioned sentences.

*5. I am afraid, but I cannot follow the meaning of Fig.6.*

Response: We will include a better explanation in the caption and make the figure more understandable.

*Minor comments for clarification:*

*Throughout the manuscript: The analysis is not done on climatological time scales, so VPD, precipitation and temperature are meteorological variables, it's not climate data.*

Response: We will change the corresponding text passages.

*l.22: I suggest to stress in the abstract already the small differences between the AutoML frameworks, eg. by writing '...AutoSklearn consistently but marginally outperformed other AutoML frameworks…'*

Response: We will change the corresponding text passages.

*l.49 and later in the manuscript: In the literature the term 'variable importance' is used with very different meanings. Please clearly state that for your work, importance refers to the contribution of a variable to model accuracy.*

Response: We will provide clarification for the use of "variable importance" in the text.

*l.49-56: I am not convinced that the conclusions of the different cited papers are strictly comparable because the analyses have been done at different temporal scales, from daily to monthly, and using different feature sets. Although the machine learning results are analysed which do not necessarily need to obey conceptual understanding, the contributions of different features are expected to differ between time scales.*

Response: We will more explicitly mention the different time scales of these studies and the limitation in comparing them.

*l.66 (and later as well, eg l.146, 149, 319, 325): Could you clarify/ give examples of what is meant by 'pipeline creation' and 'data processing steps'? The legend of Fig.A2 is hardly understandable for the non-expert without any further context or info.*

Response: The term 'pipeline' refers to the entire process of developing and training a machine learning (ML) model. A pipeline typically consists of several tasks, such as preprocessing, feature engineering, model training, hyperparameter tuning, and model deployment. Preprocessing involves various tasks to convert raw input data into a shape accessible for ML training. It typically includes steps such as data cleaning, transformation, integration, or reduction with the goal of improving the quality, accuracy, and reliability of ML models. We will provide further clarification in the corresponding text passages.

*l.81: 'predictive contribution' to what? To prediction accuracy?*

Response: We will include further clarification in the text passages.

*l. 202: Is there a reason for leaving out the VIs?*

Response: Including the VIs in the RS minimal set did not improve the prediction. Hence, we did not include them in the other feature sets. We will clarify this in the text.

*l. 232: So you compute a linear trend also for time series of just 2 years?*

Response: We will change the threshold to a longer period (5 years) and update the corresponding figures and text passages to ensure a more robust trend estimation.

*l. 241: What value does the critical difference take?*

Response: The critical difference is calculated with

$$\mathrm{CD} = q_\alpha \sqrt{\frac{k(k+1)}{6N}}$$

(CD: critical difference, q: critical values, k: number of algorithms, N: number of datasets). For more information, see Demšar (2006). We will include more clarifying information in the text.

Reference:

Demšar, J.: Statistical Comparisons of Classifiers over Multiple Data Sets, Journal of Machine Learning Research, 7, 1–30, 2006.

*Section 3.4: So the main take-away is that the patterns from AutoML in general make sense when compared to other upscaling products? Or do you want to convey another message?*

Response: We will include a concluding sentence to highlight this finding.

***l.465: the deforestation is mentioned the first time here and I cannot follow what is meant.***

Response: We will leave this part out since it is confusing and not connected to the main message of the manuscript.

***l.519-525: This last part may be slightly overstating, I do not see very clear indications of more robust and accurate GPP predictions yet.***

Response: We will adapt this part.

---

## Author Comment (AC2)

**Responses to referee #2**

*Gaber et al. test the ability of multiple automated machine learning (AutoML) approaches, each based on multiple individual machine learning methods, to upscale gross primary production (GPP) with remote sensing. They specifically test three different AutoML methods (as well as a random forest model as a baseline) with different subsets of remote sensing and meteorological data, finding that they provide very similar performance, with r^2 ranging from ~0.7-0.75 at monthly scale. They also find similar abilities to capture trends, spatial variation, and seasonality across most approaches but that none of them is particularly effective at capture monthly GPP anomalies. The best models were typically based on a combination of MODIS surface reflectance with additional remote sensing-based estimates of LAI/FPAR, land surface temperature, soil moisture, evapotranspiration, and solar-induced fluorescence (SIF); adding meteorological reanalyses of precipitation, temperature, and vapor pressure deficit did not notably improve model performance.*

*Overall, the manuscript presents an interesting comparison of some cutting edge approaches to automated machine learning and adds a new dimension to ongoing discussions of flux upscaling. It's also a well written and well-constructed study. The fact that the approaches achieve similar results to each other and to other upscaled products is itself interesting and perhaps suggests that further improvement in upscaled GPP estimates may come from avenues aside from just algorithmic optimization (e.g., better and more extensive ground data, improved remotely sensed data streams). I have a few suggestions for improved presentation and additional analysis, but overall I think this is likely to be a high quality contribution.*

Response: Thank you for your constructive feedback! Your comments are very valuable to us and will considerably contribute to improving the manuscript. We will highlight the similar results of the different frameworks and other upscaled products more in the text.

*1) My main suggestion for the analysis would be provide, if possible, a more refined and specific assessment of the importance of individual variables. The analysis of the different subsets is interesting, but I think the impact of the study could be enhanced by assessing specifically which variables within those subsets are giving the most "bang for the buck." I know random forests, for example, provide variable importance metrics and perhaps those are doable from the AutoML approaches as well? I'm curious, for example, in the RS subsets, which variables added the most predictive skill beyond what was achieved with RSmin? How important were LST and soil moisture? Did the ET and SIF data, which are themselves modeled from remote sensing data, add any additional independent information? The CSIF product, for example, is itself an upscaled SIF product based on machine learning of MODIS NBAR data, so it seems like it wouldn't necessarily add anything beyond what the methods were able to get directly from the NBAR data.*

Response: Thank you for raising this interesting point. We agree that the importance of the individual predictor variables would, indeed, add value to the study. We will include an assessment of the importance of individual variables in the form of an ablation study for the best-performing model-variable combination, AutoSklearn-RS. This can be done by calculating the permutation importance, which would indicate the model's sensitivity towards individual features. That technique takes a fitted model and has it predict on data, where one feature is recursively replaced by random noise, resulting in a potential decrease in the performance metric. The magnitude of the decrease indicates the importance of that feature to the particular model. While this technique allows us to assess the model-specific sensitivity, it can only provide a limited insight into the intrinsic information content of the input variables.

*2) I think the Discussion could use a little improvement in places. I think it would be especially helpful to improve how the findings are contextualized in light of previous literature. I'll provide more specific suggestions below.*

Response: Thank you for the suggestions. See below for the responses.

*3) I find Fig. 6 very difficult to interpret. Is it possible to present those results in a more intuitive form?*

Response: We will include a better explanation in the caption and make the figure more understandable.

*Specific comments:*

*L12: should that be "scale" instead of "scales"?*

Response: We will adapt the text.

*L14: parameterization is misspelled (missing an "e")*

Response: We will adapt the text.

*Fig. 2: Just to clarify, this is showing number of sites, not site-years, correct? If so, I wonder if it would be more relevant to show site-years since that's a better representation of how much training data is available in each biome?*

Response: We will include this information in the figure.

*L122: I think it would be worth expanding more on these different sources, including references. Especially since some of these (ET and SIF) are themselves modeled based on remote sensing. Given that, what would you expect them to add beyond what would be coming from the NBAR data itself? Would they actually be providing independent information?*

Response: We will include further discussion about the sources of the variable input. We don't expect additional information from SIF, as mentioned in your comment. ET, i.e., the ALEXI model, is derived based on energy balance and surface temperature, which is highly coupled with GPP due to stomatal control. Therefore, we hypothesize that the physical mechanisms inherent in the ET data may contribute additional information on GPP beyond remote sensing signals. We expect the feature importance analysis to shed light on the unique contribution of these variables. The revised manuscript will also discuss the impacts of modeled vs. observational variables.

*L274-275: These may be "statistically different," but to me, it seems like an r^2 of say 0.74 is not particularly different from an r^2 of 0.75 in any meaningful sense. The authors do a good job stating this later in the paper, but I do think it's worth not overinterpreting small differences even if they are "statistically significant." Any difference, however small, could be "significant" given a large enough sample size, but that doesn't necessarily make it a meaningful difference.*

Response: Thanks for raising this point. We will adapt the corresponding text passages.

*L286-297 (but also in other places throughout the results): There are places here that could use references to specific figures or panels within figures. Sometimes it's hard to tell where the results as described are shown in the figures.*

Response: We will adapt the text.

*L304-305: The overestimation of low values and underestimation of high values is interesting and consistent (I think) with some of the early studies of MODIS GPP (perhaps from David Turner and/or Faith Ann Heinsch, if I'm remembering correctly?). Some reference to those earlier works here would provide valuable context. The fact that we're still trying to solve long-standing problems is itself interesting!*

Response: Thanks for providing these insights and references. We will consider them in the text.

*L390-399: This paragraph (about differences among approaches) seems to slightly contradict the previous one (about how there aren't really major differences). I'm not suggesting that the authors do a complete rewrite of the paragraph or anything, but I do think it might be worth making sure that they are sending a consistent message: that the differences are generally pretty slight.*

Response: We will adapt the text passage.

*L401-407: It could also be that the quality of the eddy covariance data itself is a limiting factor. EC GPP is used as the ground truth in this case, but it's not a perfect representation of GPP: EC data has sources of noise and EC GPP is a modeled quantity from the more directly measured NEE. I imagine there may therefore be upper limits to the performance metrics that we can expect when upscaling EC GPP just because of uncertainties in what we're using as "truth."*

Response: We agree and will include discussions about the uncertainties and modeling background of GPP in the text.

*Section 4.2: I think this section would definitely benefit from a more thorough dive into the variable importance, as suggested in general comments. Also, I don't think there's any mention of SIF in this section while other variables composing the RS subset are discussed?*

Response: We will include an assessment of variable importance (see above) and consider the results in this paragraph.

*L433-439: The authors mention this at the end of the paragraph, but I think it could be more up front: reanalysis data (especially for precip) can be very flawed. So maybe temperature and VPD do matter (precipitation probably less so since soil moisture is already included in the model and ultimately it's soil moisture, not precipitation, that gets directly used by plants) but the reanalysis data just doesn't do a good job capturing it. Could also be worth a citation to previous literature that has assessed reanalysis data.*

Response: This is a good point. We will discuss the impact of reanalysis data with reference to previous studies, e.g., Tramontana et al. (2016). Additionally, microwave soil moisture retrievals are noisy with limitations, which may undermine their contributions to the model. Thus, the lagged precipitation may still provide useful information. Our feature importance analysis will provide further information in this respect.

Reference:

Tramontana, G., Ichii, K., Camps-Valls, G., Tomelleri, E., Papale, D., 2015. Uncertainty analysis of gross primary production upscaling using Random Forests, remote sensing and eddy covariance data. Remote Sensing of Environment 168, 360–373. https://doi.org/10.1016/j.rse.2015.07.015

*L444: I'd suggest rephrasing "It is to be explored." That's somewhat awkward, passive phrasing.*

Response: We will adapt the text.

*L463-466: This paragraph is kind of light on citations and the final sentence feels out of place and incomplete, like there's something more that should be coming that connects the first part of the paragraph to this final thought.*

Response: We will provide more references for this paragraph and embed the last sentence better in the paragraph.

*L477-484: This paragraph is also pretty light on citations. A couple suggestions: Smith et al. 2019 (Remote Sensing of Environment) on challenges specifically in dry regions and the early MODIS papers by Turner that assessed biome differences in MODIS GPP performance. It'd be interested to see the results here contextualized with the challenges that have faced remote sensing of productivity for a long time!*

Response: Thanks for suggesting these references! We will provide more references in this paragraph.

*L481: It's unclear what's meant by "high proportion of biomass" or how that would affect productivity estimation. To me, it seems like it's not high biomass that would lead to good performance but rather high seasonal variation in leaf area (which both DBF and MF have).*

Response: We will rephrase this paragraph.

*L484: A little unclear what's meant by "complex biophysical and environmental characteristics." I think it'd be worth expanding on this and being more specific.*

Response: We will rephrase this paragraph.

*L487: I think "It is to further research to…" is also somewhat awkward and passive phrasing and would suggest rewording.*

Response: We will adapt the text.

*L490: This is another good place to cite Smith et al. 2019, which also shows that drylands are underrepresented in flux networks relative to their global proportion. Haughton et al. 2018 (Biogeosciences) could be a good one too since they showed that drylands are more "unique" (meaning less easy to apply a globally-trained model to an unseen site) than most other systems, which may be partly why the underrepresentation of dryland sites in flux networks can be such a problem for upscaling in those regions.*

Response: Thank you for providing these references. We will consider them in the text.

*L504: For the Conclusions section, it might be worth expanding on what's meant by "RS" here. That's referring to a specific subset of the variables but for readers who are skimming and skip to the conclusions section, they might miss what that subset refers to.*

Response: We will adapt the text.

*L519-520: Maybe to some extent, but it's interesting to note that RF (not automated and with, I think, some amount of subjectivity in choices) performed nearly as well as the AutoML methods.*

Response: It is an interesting point. We will include this in the text.

---

## Author Comment (AC3)

Responses to community comment #1

*Gaber et al. present a comprehensive evaluation of using automated ML (AutoML) to estimate and upscale ecosystem GPP using four sets of remote sensing and reanalysis products. The comparative analysis of three AutoML frameworks reveals that AutoSklearn consistently outperforms the other frameworks and a baseline Random Forest model in reproducing spatial patterns, temporal variability, and trends in the observed GPP. Notably, the use of higher-resolution remote sensing products further enhances model performance, attributed to footprint matching. Additionally, the authors have produced a global wall-to-wall map of GPP (monthly, 0.05 deg) using AutoSklearn and a suite of remote sensing predictors, which agrees well with two other ML-based global GPP products.*

*The study highlights the potential of AutoML in quantifying global GPP, capturing its temporal and spatial variability and trend, and provides insights into feature selection for monthly GPP estimation. This topic matches the interests of the readers of Biogeosciences. While the manuscript is exceptionally well-written and the implementation of ML models is robust, several notable concerns, particularly regarding model interpretability, feature selection, and sources of uncertainty, warrant additional exploration and discussion.*

Response: Thank you, Dr. Jiangong Liu, for your valuable and constructive comments on the manuscript. Your suggestions are very helpful for us in improving the manuscript.

*Major comments:*

*1. When comparing estimations derived from "RS" and "RS + meteo", and observing no substantial improvement in model performance with additional meteorological predictors, the assertion that this is because meteorological data contains no additional information or the reanalysis data quality is not good might need further exploration (Lines 435-440). Given that several predictors from "RS + meteo" might contain overlapping information on a monthly scale (e.g., VIs, LAI, SIF, ET, and meteorological data), it might be premature to conclude that the inclusion of meteorological data yields marginal enhancement in modeling monthly GPP.*

Response: You are right that the predictors are likely to contain overlapping information at a monthly scale, and thus, the apparent results from comparing "RS" and "RS+meteo" potentially undermine the actual contribution of meteorological factors to GPP prediction. We aimed to interpret this result in the context of the overall model predictive performance measured by goodness-of-fit metrics. Thus, we will adapt the corresponding text and emphasize that the reanalysis data does not additionally improve the predictive accuracy since meteorological data largely contains overlapping information with the RS variables. We will further underscore that metrological conditions are themselves important controls of GPP in the context of literature.

*2. I am puzzled by the decision to leave out radiation (BESS_Rad) in the 'RS meteo' (Figure 3) and curious about the thinking behind splitting data sources into remote sensing and reanalysis, instead of classifying them into physical (BESS_Rad, ESA CCI, MODIS LST, and ERA5-Land) and biological (MODIS VI/LAI, CSIF, and ALEXI ET) controls. Also, I think it would be worthwhile to discuss whether SIF should be included as a predictor since it is commonly used as a GPP proxy.*

Response: BESS_Rad is part of the RS meteo variable set, as stated in Figure 3 ("Features of RS + ERA-5 Land"). We will clarify this point in the text. Splitting the data into physical and biological controls is an interesting approach and would certainly give another valuable angle at variable importance. However, it isn't easy to draw the boundaries between these categories (for instance, LST and soil

moisture are significantly influenced by biological controls). In this regard, we will perform an additional analysis to assess the feature importance of individual variables based on a permutation approach. We expect the result to comprehensively quantify the importance of variables and the relative contribution of physical and biological controls.

*3. While the Discussion does touch on various potential sources of uncertainties (e.g., section 4.2), it seems to overlook the potential for bias inherent in the eddy covariance GPP. The authors used night-time partitioned GPP, relying quite a bit on a temperature dependency function of night-time NEE. But there is still some debate about whether this dependency is exponential (Chen et al., 2023), if it can be extrapolated to the daytime (Keenan et al., 2019), and whether it should be referenced to air or soil temperature (Wohlfahrt & Galvagno, 2017). Given that AutoML isn't the easiest to interpret (Line 330), I am wondering if its top-notch performance is partly because it is picking up on some error structures during NEE partitioning.*

Response: Thank you for raising this relevant point. We will explain the origin of the GPP estimates and how they can affect prediction performance/uncertainty better in the text. Thank you also for providing the references, which we will consider in the text.

*4. I am excited about a new global GPP product. Would the authors like to give it an official name, and give the name a spotlight in the Title or Abstract? Additionally, it is recommended that the authors articulate both the interannual variability and the annual magnitude of GPP relative to the new product, as such information would likely be invaluable to the flux community. I am also curious about why the authors did not use the high-resolution RS data (500 m) for the product, considering it seems to pull better performance.*

Response: Our analysis focuses primarily on the benchmark of different AutoML frameworks. The upscaled maps were mainly used to verify the results of AutoML in comparison to benchmarking products. At this stage, the release of a new GPP dataset is not planned but could be considered in the future. 500m RS data improved the performance significantly; we used the 0.05 degree data for upscaling to compare with other upscaled datasets, which are typically at a similar or coarser resolution. In light of our result, the production of 500m-resolution data from upscaling is highly encouraged to improve accuracy and reduce uncertainties associated with scaling errors. We will clarify and highlight these aspects in the discussion section.

*Minor comments*

*Line 90: Since negative outliers are in a unit of "gC m-2 d-1", did the authors aggregate daily values to monthly for both fluxes and their predictors? More details should be provided for the quality control.*

Response: We used the monthly data provided by the original data sources, i.e., FLUXNET2015, AmeriFlux ONEFLUX, and ICOS. The monthly data has been aggregated from daily and half-hourly/hourly values. Outlier removal was performed on monthly data that corresponds to average daily NEE. We will provide clarification in the text.

*Line 100: Add the source/reference for IGBP here, and also in Fig 2.*

Response: We will include this in the text.

*Line 115: It is a very minor point, but I think terminology for explanatory variables/predictor (e.g., Table 1)/feature (e.g., line 40) is used a bit random in the manuscript. Though they share the same meaning, readers might get confused.*

Response: We will better align the terminology.

**Line 130-140: It might be worthwhile to relocate this paragraph concerning the challenges with CASH to the Introduction to serve as an additional motivation statement. In the current Introduction, the authors highlighted the advantages of using AutoML, which are "... to overcome the challenges of algorithm selection, hyperparameter tuning, and pipeline creation through an automated approach". They introduced well the existing problem of feature selection. However, the knowledge gaps in the existing ML-based products of fluxes regarding algorithm selection and hyperparameter tuning should also be clarified.**

Response: Thank you for raising this point. We will include the challenge of algorithm selection and hyperparameter tuning in current ML-based products in the introduction.

**Line 255: Offering details about the calculation of trends, seasonality, across-site variability, and anomalies in the Methodology section, prior to Figure 10, might enhance comprehension. I am also unsure what R2 values mean for trend comparison, as trends are the fitted slopes.**

Response: We will include a reference to 2.3.2 to clarify the calculation of trends, seasonality, across-site variability, and anomalies. The R2 for trends represents the spatial variability of their slopes. We will clarify that in the text.

**Figure 7: what do R2 values smaller than -1 mean?**

Response: We define R2 as the coefficient of determination that provides a measure of the proportion of variation that can be predicted from the predictor variables. Negative R2 values mean that the model performs worse than a simple model that just predicts the mean of the dependent variable. This definition of R2 aligns with the Nash- Nash–Sutcliffe model efficiency coefficient that is typically used in hydrological models, and it is commonly used as a metric for regression models in machine learning applications. We will provide more descriptions in the text to improve clarity.

**Line 490: While the models also underestimate large GPP values (Line 305), further discussion on this aspect may provide additional insight.**

Response: We will include further discussion/literature regarding this behavior.

**Line 520: I appreciate the authors raising this point about the cautious use of AutoML. The inherently 'black-box' nature of AutoML, which presents challenges in interpretability as indicated (Line 330), is a notable issue.**

Response: Thanks for this feedback!

---

## Author Response (AR1)

**Point-to-point reply**

**Referee #1**

| Referee comment | Authors' response | Authors' changes |
|---|---|---|
| What is the reason for doing this analysis at a monthly temporal scale when structural vegetation changes dominate rather than finer temporal resolution? I would expect higher gains from AutoML and also more differentiated contributions between predictor variables (especially meteorological features) at higher temporal resolution. This is also the time scale which is more relevant to be able to properly represent seasonal and anomalous trajectories. I would expect large potential from automated model tuning especially for short extreme events, which are relevant for the carbon uptake and hard to represent in a data-driven model set-up, but clearly smeared out at a monthly time step. Much of the discussion in section 4.2 does neglect the coarse time step when for example LUE changes are not expected to play major role. | The temporal resolution is indeed an important factor in the contribution of the different predictor variables. A higher temporal resolution could enable the models to represent better anomalies, extreme events, and their impact on GPP (see, e.g. Bodesheim et al. (2018)). Since many previous upscaling works focus on monthly scales, and these data have been instrumental in informing global long-term dynamics of GPP across different regions in many studies, we have chosen to perform this evaluation at monthly scales as an initial step. Our team has follow-up studies that examined more advanced machine learning algorithms, such as the temporal fusion transformer (TFT), in modeling the dynamics of GPP at hourly scales across space (Rumi Nakagawa et al., 2023). More assessments are necessary to quantify machine learning performance under different time scales. We will make sure to highlight better the consideration of temporal scales on the upscaling framework and model choice in our discussion in the revised manuscript. | We highlighted the temporal aspect more prominently and included a brief discussion about the implications of monthly data on our benchmark and variable importance results.

**l. 541:** *"The ability of the frameworks to reproduce GPP patterns and the corresponding variable importance must be evaluated in light of the choice of temporal resolution. In this study, we evaluated machine learning upscaling of monthly GPP dynamics, which are dominated by light availabilities and seasonal changes in vegetation structures. However, at shorter time scales, such as hourly or daily, GPP is more closely aligned with diurnal and short-term variations in meteorological conditions such as temperature and VPD. Thus, these variables are likely more influential in predicting GPP at these higher frequencies (Frank et al., 2015; von Buttlar et al., 2018). Additionally, complex machine learning models may also offer greater benefits at harnessing the large data quantities involved in predicting GPP at hourly or daily scales. Further research is needed to benchmark machine learning algorithms and assess choices of environmental data in predicting GPP across different timescales."* |

| A number of predictor variables are model outputs themselves, relying on input data and model assumptions. This is not discussed at all. | We will include further discussion about the sources of the variable input with a focus on introduced uncertainty from the modeling process. | **l. 497:** *"Furthermore, it may depend on the choice of temporal and spatial scales and data quality, given that many of the input features are themselves model outputs."* |
|---|---|---|
| What is the reason for ingesting both SIF and instantaneous SIF, or both PAR and RSDN? | Our approach was to include as many predictor variables as possible and let the AutoML frameworks identify what variables are necessary for a good prediction. This includes variables showing a high intercorrelation and potentially small differences in predictive capacity. We will provide clarification in the revised manuscript. | We highlighted this goal more clearly in section 2.1.2

**l. 122:** *"Our goal was to provide as many explanatory variables as possible and let the frameworks decide which to use."* |
| How is the temporal aggregation done? | We aggregated with a simple average within the respective period after filling the data gaps (see below). We will clarify this in the text. | We included a more elaborate explanation in section 2.1.2

**l. 137:** *"All datasets were resampled to a 0.05 ° spatial resolution, and data gaps were filled at the native temporal resolution before resampling to a monthly frequency using a simple average."* |
| How do you handle data gaps? | We filled gaps at native temporal resolution. For high-resolution data products (frequency <=4 days), such as NBAR, LAI/FPAR, BESS, CSIF, and CCI, we filled gaps less or equal to 5 days (8 days for products with a 4 day resolution) with the average of a 15-day moving window. We gap-filled LST with a 9-day moving window since we observed higher variations. Soil moisture was filled after Walther et al. (2021) with moving window medians for short gaps and mean seasonal cycle for long gaps. We will clarify this in the text. | We included the information about gap-filling in section 2.1.2

**l. 138:** *"We performed the gap filling as follows: We filled gaps of less or equal five days (8 days for four-day resolution datasets) with the average of a fifteen-days moving window for high-frequency datasets (NBAR, LAI, FPAR, BESS_Rad, CSIF). We gap-filled LST with a 9-day moving window because we observed higher variations. For SM, we followed Walther et al. (2022) and used the moving window median for short gaps and the mean seasonal cycle for long gaps."* |

| | | |
|---|---|---|
| Handling of bad data quality is only mentioned for the site-level fluxes, what about the explanatory variables? | We used NBAR, where >75% high-resolution NBAR pixels were available from full BRDF inversion. We applied the quality control mask for LST where the average emissivity error is < 0.02. LAI/FPAR was used with and without saturation. We used all data for soil moisture. We will include this in the text. | We included the information on our handling of bad data quality in section 2.1.2

**l. 134:** *"We filtered the data for poor-quality pixels, performed gap-filling, and matched spatial and temporal resolutions. We used NBAR, where more than 75 % of high-resolution NBAR pixels were available from the full BRDF inversion. We applied the quality control mask for LST, where the average emissivity error was less than 0.02. LAI and FPAR were used with and without saturation. All datasets were resampled to a 0.05 ° spatial resolution, and data gaps were filled at the native temporal resolution before resampling to a monthly frequency using a simple average."* |
| Specify more clearly the data sources, e.g. for the CCI soil moisture, which version did you use? Presumably, FluxCom v6 refers to the FluxCom set up with RSonly ( only satellite-based predictors using MODIS collection 6), which is 8-daily and at high spatial resolution? | We used CCI Soil moisture v.06.1 and FluxCom v6 RS only. We will include this in the text. | We mentioned the dataset versions in several parts of the manuscript |
| Spatial resolution: Why not also ingest tower meteorology instead of the coarser ERA5-Land? The scale mismatch could be further discussed, especially between a 0.05deg pixel and the tower footprint. The way the authors approach the analysis suggests using the 0.05deg pixel is the generally accepted default, which is not the case. | Thank you for raising this interesting point. The spatial mismatch is a large uncertainty factor in the prediction, as outlined in the manuscript (l.309-314). Tower meteorology is expected to increase predictive performance substantially compared to the coarse-resolution ERA-5 product. Regarding using meteorological variables as predictor variables for global upscaling, however, tower meteorology poses a limitation due to its spatially constrained availability. It cannot be used as a predictor for regions | We highlighted the use of independent explanatory variables in section 1

**l. 41:** *"These ML models use independent globally available explanatory data from remote sensing or other continuous model outputs to infer a functional relationship to the GPP measurements, which can be used to predict GPP in areas beyond the limited flux tower footprints"* |

| | | |
|---|---|---|
| | where no flux tower data exists. For this reason, we chose ERA-5 land, since it is globally available and, hence, can be used for global predictions. It would be interesting to evaluate uncertainties in reanalysis data using tower meteorology and understand the potential impacts on upscaling uncertainties. We will clarify and discuss this aspect in the text. | |
| In parts the manuscript uses very technical language and describes key concepts only in a very short manner. I suggest to rephrase certain passages to make the manuscript better accessible to a wider audience which may also not be very familiar with the newest developments in the machine learning world – or at least expand more in the supporting information. Examples of very technical sentences in my opinion are l.160-161, l.165-166, l.170-172, l.177-181, l.243-246 | We will make the text more accessible and reformulate the mentioned sentences. | We rephrased the technical sentences

**l. 187:** *"The meta-learner uses knowledge from previous experiments with similar datasets and can, therefore, select promising ML models to start with instead of training from scratch each time."*

**l. 191:** *"H2O AutoML draws from a set of base models, which, in the developer's terminology, are divided into the model families of Gradient Boosting models (GBM), XGBoost GBMs, GLMs, a default Random Forest model (DRF), Extremely Randomized Trees (XRT), and feed-forward neural networks. The framework trains these models in a predefined order with increasing diversity and complexity, using pre-specified hyperparameters or tuning them by random search."*

**L. 195:** *"In addition to the individual base models, H2O AutoML creates ensembles of the base models, combining their predictions through a generalized linear model (GLM) by default. The ensembles consist of either all base models or only the best-performing base models from each model family. H2O AutoML then ranks the performance of individual models and model ensembles using an internal cross-validation (CV). The best-performing model is used for prediction."* |

| | | |
|---|---|---|
| | | **L. 204:** *"These models are combined in a multi-layer stack ensembling process: AutoGluon first generates predictions from each base model. The predictions are then concatenated with the original features and passed to another set of models (the stacker models) in the next layer. Their predictions can be concatenated again and passed to the next layer, and so on, creating a layered structure of model sets and concatenation steps. The predictions of the last layer are combined in an ensemble selection step (Caruana et al., 2004). Each layer consists of the same base model types and hyperparameters."* |
| I am afraid, but I cannot follow the meaning of Fig.6. | We will include a better explanation in the caption and make the figure more understandable. | We included more supportive graphical elements and a more extensive caption

**l. 317:** *"Figure 6 Critical difference (CD) diagrams (Demšar, 2006) for the ranks of the frameworks and variable sets, which are typically used to compare the performance of multiple algorithms on multiple problems (in this case, repeated cross-validations). The graphs rank the performance of different framework-variable combinations on the x-axis, with one being the best rank. The ranks shown are the average ranks from all repeated cross-validations for each of the frameworks/variable sets. The performance (r2) is given for predicting total GPP and for its different spatial and temporal components: trend, seasonality, anomalies, and across-site variability. We evaluated whether the ranks are statistically significantly different from each other using the critical difference (CD) obtained from a Nemenyi post hoc test. If the difference between the ranks is less than the CD, we assume a nonsignificant difference in ranks, indicated by a red crossbar between the rank markers. On the left side (a), the ranks of the frameworks trained on the "RS" explanatory variables are shown. On the right side (b), the ranks of AutoSklearn trained on different sets of explanatory variables are shown."* |

| | | |
|---|---|---|
| Throughout the manuscript: The analysis is not done on climatological time scales, so VPD, precipitation and temperature are meteorological variables, it's not climate data. | We will change the corresponding text passages. | We changed the terminology throughout the text |
| l.22: I suggest to stress in the abstract already the small differences between the AutoML frameworks, eg. by writing '…AutoSklearn consistently but marginally outperformed other AutoML frameworks…' | We will change the corresponding text passages. | We changed the abstract and highlighted the result as proposed

**l. 21:** *"We found that the AutoML framework AutoSklearn consistently outperformed other AutoML frameworks as well as a classical Random Forest regressor in predicting GPP, but with small performance differences, reaching an r2 of up to 0.75."* |
| l.49 and later in the manuscript: In the literature the term 'variable importance' is used with very different meanings. Please clearly state that for your work, importance refers to the contribution of a variable to model accuracy. | We will provide clarification for the use of "variable importance" in the text. | We included an explanation in section 1

**l. 85:** *"In addition, we evaluate the variable importance, i.e., the contribution of various remotely sensed vegetation structure variables, proxies for photosynthetic activity and environmental stress (i.e., greenness, land surface temperature, soil moisture, evapotranspiration), and meteorological factors, for the performance of the AutoML frameworks."* |
| l.49-56: I am not convinced that the conclusions of the different cited papers are strictly comparable because the analyses have been done at different temporal scales, from daily to monthly, and using different feature sets. Although the machine learning results are analysed which do not necessarily need to obey conceptual understanding, the contributions of different features are expected to differ between time scales. | We will more explicitly mention the different time scales of these studies and the limitation in comparing them. | We mentioned the time scales for each of the cited papers |
| l.66 (and later as well, eg l.146, 149, 319, 325): Could you clarify/ give examples of what is meant by 'pipeline creation' and | The term 'pipeline' refers to the entire process of developing and training a machine learning (ML) model. A pipeline | We included additional explanations and improved the caption of the figure |

| | | |
|---|---|---|
| 'data processing steps'? The legend of Fig.A2 is hardly understandable for the non-expert without any further context or info. | typically consists of several tasks, such as preprocessing, feature engineering, model training, hyperparameter tuning, and model deployment. Preprocessing involves various tasks to convert raw input data into a shape accessible for ML training. It typically includes steps such as data cleaning, transformation, integration, or reduction with the goal of improving the quality, accuracy, and reliability of ML models. We will provide further clarification in the corresponding text passages. | **l. 163:** *"The pipeline refers to the entire process of developing and training an ML model and typically consists of several tasks, such as preprocessing, feature engineering, model training, hyperparameter tuning, and model deployment."*

**l. 178:** *"AutoML draws from a pool of classical ML algorithms (base models) and preprocessing methods and selects or combines the most appropriate candidates for the ML problem. Typically, AutoML frameworks create model ensembles by combining the predictions of their base models, either through a simple aggregation or through yet another model that uses the predictions of the base models as input features. This approach is often superior to individual predictions because it can overcome the limitations of the individual base models (van der Laan et al., 2007)."*

**l. 666:** *"Figure A5 Detailed use of preprocessing algorithms by AutoSklearn. The chart shows the distribution of the mean RMSE for each base model type across all folds within each repetition of the cross-validation. We considered only the best-performing models for each model class within each fold. The RMSE is min-max scaled from zero to one within each cross-validation fold to account for variations in the data's predictability depending on the data's split. The use of preprocessing algorithms is shown as colors in the proportions of their usage in each bin."* |
| l.81: 'predictive contribution' to what? To prediction accuracy? | We will include further clarification in the text passages. | We adapted the corresponding text passage

**l. 85:** *"In addition, we evaluate the variable importance, i.e., the contribution of various remotely sensed vegetation structure variables, proxies for photosynthetic activity and environmental stress (i.e., greenness, land surface temperature, soil moisture, evapotranspiration), and meteorological factors, for the performance of the AutoML frameworks."* |

| | | |
|---|---|---|
| l. 202: Is there a reason for leaving out the VIs? | Including the VIs in the RS minimal set did not improve the prediction. Hence, we did not include them in the other feature sets. We will clarify this in the text. | We provided a clarifying sentence

**l. 231:** *"As we did not detect any further significant performance improvements by including VIs, we did not consider them in other variable sets."* |
| l. 232: So you compute a linear trend also for time series of just 2 years? | We will change the threshold to a longer period (5 years) and update the corresponding figures and text passages to ensure a more robust trend estimation. | We changed the analysis and considered only trends where time series of minimum 5 years were available. We detrended anomalies only if this requirement was satisfied. We changed the corresponding benchmark metrics and graphs (Fig. 4, 5, 6, 8, and 9).

**l. 259:** *"In addition to obtaining performance metrics for the total time series prediction, we decomposed the time series to evaluate the performance in different spatial and temporal domains. We computed the components as follows: we obtained trends by linear regression of the entire time series (using the slope for evaluation with RMSE and r2), seasonality (mean seasonal cycle) by month-wise averaging, and anomalies as their residuals after detrending and removing seasonality. Furthermore, we calculated an across-site variability from the multi-year mean at each site. For this analysis, we considered only sites with a minimum of 24 months of measurements to minimize the error from sites with just a few measurements, leaving us with 211 sites. When calculating trend metrics, we only considered sites with at least 60 months of measurements for our trend evaluations. Time series anomalies were detrended only when this minimum was reached; otherwise, we simply removed the seasonal component from the time series."* |
| l. 241: What value does the critical difference take? | The critical difference is calculated with $CD=q\_\alpha \sqrt{k(k+1)/6N}$
(CD: critical difference, q: critical values, k: number of algorithms, N: number of datasets). For more information, see | We provided more background on the CD. An extensive explanation can be found in Demšar (2006).

**l. 300:** *"We rejected the null hypothesis (no significant difference between the two frameworks) if the difference between the* |

| | Demšar (2006). We will include more clarifying information in the text. | *average ranks exceeded a critical difference (CD)), which depends on the critical value of the Studentized range distribution (Demšar, 2006)."* |
| --- | --- | --- |
| Section 3.4: So the main take-away is that the patterns from AutoML in general make sense when compared to other upscaling products? Or do you want to convey another message? | We will include a concluding sentence to highlight this finding. | We added a concluding sentence to clarify our message

**l. 485:** *"Thus, AutoSklearn shows good agreement with the GPP patterns predicted by FluxSat, whereas it deviates more strongly from the FluxCom product."* |
| l.465: the deforestation is mentioned the first time here and I cannot follow what is meant. | We will leave this part out since it is confusing and not connected to the main message of the manuscript. | We removed this sentence |
| l.519-525: This last part may be slightly overstating, I do not see very clear indications of more robust and accurate GPP predictions yet. | We will adapt this part. | We formulated this part less overstating

**l. 640:** *"In addition, AutoML enables the exploration of a wide range of models and algorithms, uncovering potential relationships and patterns that may have been missed manually. However, we were unable to demonstrate that AutoML produces GPP predictions that are considerably more accurate and robust than classical ML models. In particular, the non-automated Random Forest model performed almost as well as AutoSklean."* |

**Referee #2**

| Referee comment | Author's response | Author's changes |
| --- | --- | --- |
| My main suggestion for the analysis would be provide, if possible, a more refined and specific assessment of the importance of individual variables. The analysis of the different subsets is interesting, but I think the impact of the study could be enhanced by assessing specifically which variables within those subsets are giving | Thank you for raising this interesting point. We agree that the importance of the individual predictor variables would add value to the study. We will include an assessment of the importance of individual variables in the form of an ablation study for the best-performing model-variable combination, AutoSklearn- | We included a variable importance analysis for RS minimal, RS, and RS meteo. The results are presented in section 3.1 and discussed in section 4.2. We included additional background information in the appendix. Statements in the discussion, conclusion, and abstract were slightly adapted to reflect the additional insights. We added the following new figures: 7, A1, A2, and A3. |

the most "bang for the buck." I know random forests, for example, provide variable importance metrics and perhaps those are doable from the AutoML approaches as well? I'm curious, for example, in the RS subsets, which variables added the most predictive skill beyond what was achieved with RSmin? How important were LST and soil moisture? Did the ET and SIF data, which are themselves modeled from remote sensing data, add any additional independent information? The CSIF product, for example, is itself an upscaled SIF product based on machine learning of MODIS NBAR data, so it seems like it wouldn't necessarily add anything beyond what the methods were able to get directly from the NBAR data.

RS. This can be done by calculating the permutation importance, which would indicate the model's sensitivity towards individual features. That technique takes a fitted model and has it predict on data, where one feature is recursively replaced by random noise, resulting in a potential decrease in the performance metric. The magnitude of the decrease indicates the importance of that feature to the particular model. While this technique allows us to assess the model-specific sensitivity, it can only provide a limited insight into the intrinsic information content of the input variables.

**l. 340:** *"To determine which explanatory variable was most effective for predicting GPP, we evaluated the permutation importance of the variables for the AutoSklearn framework. Permutation importance is the decrease in prediction performance on the test dataset when one of the variables is randomly shuffled to break its relationship with the target variable. To deal with collinearity among the explanatory variables (Fig. A1), we first clustered them based on their average mutual Pearson correlation coefficient, regardless of their data source or ecological function. Variables with an average correlation greater than 0.7 were clustered and permuted together, resulting in clusters focused around specific meteorological characteristics (e.g., precipitation, temperature), vegetation properties, or combinations of reflectance bands but also combining features that are not directly biophysically related (Fig. A2 and A3)."*

**l. 357:** *"Our results show the largest decrease in r2 of AutoSklean-RS when removing the cluster of SIF, LAI, and FPAR, followed by PAR, RSDN, LST, and ET (Fig. 7). The other variables do not substantially reduce the framework performance. Trained on "RS meteo," AutoSklearn's variable importance gives a similar picture despite slightly different clusters due to the inclusion of the meteorological variables. Again, the cluster of SIF, LAI, and FPAR shows by far the highest importance, followed by the PAR, RSDN, ET, and temperature-related variables (Fig. 7). The meteorological variables temperature, VPD, and precipitation are generally in clusters of lower importance, as are the MODIS NBAR features. In contrast, the "RS minimal" product shows the highest variable importance for the visible NBAR spectrum, followed by NIR and PAR in descending order. The SWIR bands are hardly used in any setup."*

**l. 525:** *"The permutation importance of explanatory variables provides further insight into which variables AutoSklearn uses and which are indifferent to the framework. Our results show that both "RS" and "RS meteo"-trained AutoSklearn frameworks rely primarily on features of canopy structure (LAI, FPAR), proxies for photosynthetic activity (SIF), and ET, which strongly couples with GPP in favorable environmental conditions. Meteorological information, such as temperature and VPD, are less relevant for the model prediction. This suggests that the insignificant changes in performance between "RS" and "RS meteo" may be related to a small additional contribution of meteorological conditions to the prediction of monthly GPP beyond what is already provided by vegetation structure and PAR. Soil moisture was also found to have minimal influence overall, which might be partly due to uncertainties and noises in the remote sensing soil moisture data and due to its coarse spatial resolution. It is also important to note that previous studies have demonstrated the importance of soil moisture from SMAP in predicting GPP in water-limited ecosystems (Dannenberg et al., 2023; Kannenberg et al., 2024). The performance difference between "RS minimal" (NBAR and PAR only) and "RS" variables seems to be driven at least partly by features that are themselves model outputs based on MODIS NBAR, i.e., SIF, LAI, and FPAR. We grouped the variables into clusters with high correlation to improve the interpretability of the importance measures. However, we could not completely eliminate correlations between clusters. High correlations between, for example, PAR and LST, and ET and PAR, as well as lower correlations between other variables, could not be taken into account and introduced further uncertainty in the reported variable importance. The ability of the frameworks to reproduce GPP patterns and the corresponding variable importance must be evaluated in light of the choice of temporal resolution. In this study, we evaluated machine learning upscaling of monthly GPP*

| | | |
|---|---|---|
| | | *dynamics, which are dominated by light availabilities and seasonal changes in vegetation structures. However, at shorter time scales, such as hourly or daily, GPP is more closely aligned with diurnal and short-term variations in meteorological conditions such as temperature and VPD. Thus, these variables are likely more influential in predicting GPP at these higher frequencies (Frank et al., 2015; von Buttlar et al., 2018). Additionally, complex machine learning models may also offer greater benefits at harnessing the large data quantities involved in predicting GPP at hourly or daily scales. Further research is needed to benchmark machine learning algorithms and assess choices of environmental data in predicting GPP across different timescales."* |
| I find Fig. 6 very difficult to interpret. Is it possible to present those results in a more intuitive form? | We will include a better explanation in the caption and make the figure more understandable. | We included more supportive graphical elements and a more extensive caption

**l. 317:** *"Figure 6 Critical difference (CD) diagrams (Demšar, 2006) for the ranks of the frameworks and variable sets, which are typically used to compare the performance of multiple algorithms on multiple problems (in this case, repeated cross-validations). The graphs rank the performance of different framework-variable combinations on the x-axis, with one being the best rank. The ranks shown are the average ranks from all repeated cross-validations for each of the frameworks/variable sets. The performance (r2) is given for predicting total GPP and for its different spatial and temporal components: trend, seasonality, anomalies, and across-site variability. We evaluated whether the ranks are statistically significantly different from each other using the critical difference (CD) obtained from a Nemenyi post hoc test. If the difference between the ranks is less than the CD, we assume a nonsignificant difference in ranks, indicated by a red crossbar between the rank markers. On the left side (a), the ranks of the frameworks trained on the "RS" explanatory variables are shown.* |

| | | |
|---|---|---|
| | | *On the right side (b), the ranks of AutoSklearn trained on different sets of explanatory variables are shown."* |
| L12: should that be "scale" instead of "scales"? | We will adapt the text. | Changed |
| L14: parameterization is misspelled (missing an "e") | We will adapt the text. | Changed |
| Fig. 2: Just to clarify, this is showing number of sites, not site-years, correct? If so, I wonder if it would be more relevant to show site-years since that's a better representation of how much training data is available in each biome? | We will include this information in the figure. | We changed the yellow column to the number of site-months, consistent with the terminology used in the rest of the manuscript |
| L122: I think it would be worth expanding more on these different sources, including references. Especially since some of these (ET and SIF) are themselves modeled based on remote sensing. Given that, what would you expect them to add beyond what would be coming from the NBAR data itself? Would they actually be providing independent information? | We will include further discussion about the sources of the variable input. We do not expect additional information from SIF, as mentioned in your comment. ET, i.e., the ALEXI model, is derived based on energy balance and surface temperature, which is highly coupled with GPP due to stomatal control. Therefore, we hypothesize that the physical mechanisms inherent in the ET data may contribute additional information on GPP beyond remote sensing signals. We expect the feature importance analysis to shed light on the unique contribution of these variables. The revised manuscript will also discuss the impacts of modeled vs. observational variables. | We included this in the discussion about the variable importance, Also in the light of modeled and observational variables.

**l. 525:** *"The permutation importance of explanatory variables provides further insight into which variables AutoSklearn uses and which are indifferent to the framework. Our results show that both "RS" and "RS meteo"-trained AutoSklearn frameworks rely primarily on features of canopy structure (LAI, FPAR), proxies for photosynthetic activity (SIF), and ET, which strongly couples with GPP in favorable environmental conditions. Meteorological information, such as temperature and VPD, are less relevant for the model prediction. This suggests that the insignificant changes in performance between "RS" and "RS meteo" may be related to a small additional contribution of meteorological conditions to the prediction of monthly GPP beyond what is already provided by vegetation structure and PAR. Soil moisture was also found to have minimal influence overall, which might be partly due to uncertainties and noises in the remote sensing soil moisture data and due to its coarse spatial resolution. It is also important to note that previous studies have demonstrated the importance of soil moisture from SMAP in predicting GPP in water-limited* |

| | | |
|---|---|---|
| | | *ecosystems (Dannenberg et al., 2023; Kannenberg et al., 2024). The performance difference between "RS minimal" (NBAR and PAR only) and "RS" variables seems to be driven at least partly by features that are themselves model outputs based on MODIS NBAR, i.e., SIF, LAI, and FPAR. We grouped the variables into clusters with high correlation to improve the interpretability of the importance measures. However, we could not completely eliminate correlations between clusters. High correlations between, for example, PAR and LST, and ET and PAR, as well as lower correlations between other variables, could not be taken into account and introduced further uncertainty in the reported variable importance. The ability of the frameworks to reproduce GPP patterns and the corresponding variable importance must be evaluated in light of the choice of temporal resolution. In this study, we evaluated machine learning upscaling of monthly GPP dynamics, which are dominated by light availabilities and seasonal changes in vegetation structures. However, at shorter time scales, such as hourly or daily, GPP is more closely aligned with diurnal and short-term variations in meteorological conditions such as temperature and VPD. Thus, these variables are likely more influential in predicting GPP at these higher frequencies (Frank et al., 2015; von Buttlar et al., 2018). Additionally, complex machine learning models may also offer greater benefits at harnessing the large data quantities involved in predicting GPP at hourly or daily scales. Further research is needed to benchmark machine learning algorithms and assess choices of environmental data in predicting GPP across different timescales."* |
| L274-275: These may be "statistically different," but to me, it seems like an r^2 of say 0.74 is not particularly different from an r^2 of 0.75 in any meaningful sense. The authors do a good job stating this later in the paper, but I do think it's | Thanks for raising this point. We will adapt the corresponding text passages. | We added a statement, highlighting the marginal difference.

**l. 310:** *"However, their difference in performance is marginal."* |

| | | |
|---|---|---|
| worth not overinterpreting small differences even if they are "statistically significant." Any difference, however small, could be "significant" given a large enough sample size, but that doesn't necessarily make it a meaningful difference. | | |
| L286-297 (but also in other places throughout the results): There are places here that could use references to specific figures or panels within figures. Sometimes it's hard to tell where the results as described are shown in the figures. | We will adapt the text. | We included more references to the figures discussed in the text |
| L304-305: The overestimation of low values and underestimation of high values is interesting and consistent (I think) with some of the early studies of MODIS GPP (perhaps from David Turner and/or Faith Ann Heinsch, if I'm remembering correctly?). Some reference to those earlier works here would provide valuable context. The fact that we're still trying to solve long-standing problems is itself interesting! | Thanks for providing these insights and references. We will consider them in the text. | The references are mentioned in section 4.1 to put our results into perspective

**l. 469:** *"AutoSklearn trained on "RS" explanatory variables tended to overestimate small GPP values while underestimating large GPP values. This behavior was already observed in the FluxCom (RS), FluxSat, and several light use efficiency models (Yuan et al., 2014; Joiner et al., 2018). It has also been shown for the early MODIS GPP product (Running et al., 2004), where the overestimation was attributed to an artificially high FPAR while the underestimation was related to low light use efficiency in the MODIS algorithm (Turner et al., 2006). Another reason could be the strong reliance of the AutoSklearn framework on tree-based models (Fig. 10). These models are constructed by recursively partitioning the feature space into small regions to which they fit a simple model, which limits them in their ability to extrapolate beyond the range of target values already observed. Furthermore, our predictions showed differing prediction quality at the land cover level, which might result from biome-specific circumstances and the* |

| | | |
|---|---|---|
| | | *availability of measurement sites. For example, biomes with a pronounced seasonal cycle, such as DBF or MF, exhibit high overall r2, whereas EBF and WET show large variability that the model could not capture. In addition, variability within a land cover type could affect the performance assessment, such as for SH, which includes both arid and subarctic shrublands."* |
| L390-399: This paragraph (about differences among approaches) seems to slightly contradict the previous one (about how there aren't really major differences). I'm not suggesting that the authors do a complete rewrite of the paragraph or anything, but I do think it might be worth making sure that they are sending a consistent message: that the differences are generally pretty slight. | We will adapt the text passage. | We chose a less contradicting formulation

**l. 458:** *"The performance differences between the frameworks are statistically significant but slight. AutoSklearn consistently outperforms H2O AutoML, AutoGluon, and Random Forest."* |
| L401-407: It could also be that the quality of the eddy covariance data itself is a limiting factor. EC GPP is used as the ground truth in this case, but it's not a perfect representation of GPP: EC data has sources of noise and EC GPP is a modeled quantity from the more directly measured NEE. I imagine there may therefore be upper limits to the performance metrics that we can expect when upscaling EC GPP just because of uncertainties in what we're using as "truth." | We agree and will include discussions about the uncertainties and modeling background of GPP in the text. | We added an additional paragraph on the limitations of night-time partitioned GPP to section 4.4

**l. 603:** *"Finally, an additional limitation is introduced by the eddy covariance measurements themselves. We use night-time-partitioned GPP, which is modeled as the difference between NEE and ecosystem respiration. While NEE and night-time respiration are directly measurable, daytime respiration is modeled with a temperature response function, which extrapolates from night-time respiration (Reichstein et al., 2005). Up to this point, it is not conclusively clarified how reliably this approach can be employed, considering that it is indifferent to some environmental stress factors and changes in respiration behavior between day and nighttime (Wohlfahrt and Galvagno, 2017; Keenan et al., 2019; Tramontana et al., 2020). The inherent uncertainty and bias in the ground truth GPP data could be a potential cap to the performance we can obtain in our efforts to predict GPP."* |

| | | |
|---|---|---|
| Section 4.2: I think this section would definitely benefit from a more thorough dive into the variable importance, as suggested in general comments. Also, I don't think there's any mention of SIF in this section while other variables composing the RS subset are discussed? | We will include an assessment of variable importance (see above) and consider the results in this paragraph. | The importance of explanatory variables is discussed in section 4.2. We added insights into what variables were important in the light of what ecological function they represent. Due to collinearity between many of the explanatory variables, we could not separate the importance for some variable clusters. For details, see the sections on variable importance above (**l. 340, 357, 525**). |
| L433-439: The authors mention this at the end of the paragraph, but I think it could be more up front: reanalysis data (especially for precip) can be very flawed. So maybe temperature and VPD do matter (precipitation probably less so since soil moisture is already included in the model and ultimately it's soil moisture, not precipitation, that gets directly used by plants) but the reanalysis data just doesn't do a good job capturing it. Could also be worth a citation to previous literature that has assessed reanalysis data. | This is a good point. We will discuss the impact of reanalysis data with reference to previous studies, e.g., Tramontana et al. (2016). Additionally, microwave soil moisture retrievals are noisy with limitations, which may undermine their contributions to the model. Thus, the lagged precipitation may still provide useful information. Our feature importance analysis will provide further information in this respect. | We discuss the role of the ERA-5 Land data in our analysis of the variable importance.

**l. 512:** *"Including the meteorological explanatory features (ERA5-Land) in the training data does not significantly improve the prediction quality for any of the frameworks. This implies that meteorological data may not contain additional information that the machine learning frameworks in this study can effectively use to predict GPP. A possible explanation is that the "RS" set already includes variables, such as LST, ET, and soil moisture, that encode information about the instantaneous environmental stress on LUE due to adverse meteorological conditions, which are important controls of GPP (Bloomfield et al., 2023). At a monthly scale, the information contained in the meteorological data may overlap with the data provided by the "RS" variables. Furthermore, the coarse resolution of the reanalyzed meteorological data could introduce additional uncertainty due to a scale mismatch with the flux tower footprint sizes. Finally, its quality may not adequately inform the machine learning models due to the presence of large uncertainties. For example, Joiner and Yoshida (2020) showed that using site-measured meteorological data rather than reanalyzed data significantly improved the performance of GPP predictions. Further studies could potentially evaluate these uncertainties by comparing models trained with tower meteorological data to gridded reanalysis datasets."* |

| L444: I'd suggest rephrasing "It is to be explored." That's somewhat awkward, passive phrasing. | We will adapt the text. | l. 555: *"We suggest further exploring how to align the datasets better, e.g., through better representing the flux tower footprints (Xiao et al., 2008; Yu et al., 2018; Chu et al., 2021)."* |
|---|---|---|
| L463-466: This paragraph is kind of light on citations and the final sentence feels out of place and incomplete, like there's something more that should be coming that connects the first part of the paragraph to this final thought. | We will provide more references for this paragraph and embed the last sentence better in the paragraph. | We rephrased the paragraph

l. 570: *"High anomalies occurred in mainly temperate and semi-arid climates, the latter of which have also been shown to dominate the interannual variability of the global terrestrial carbon sink (Ahlström et al., 2015). Besides random variations included in the anomalies, reasons could be non-seasonal events, such as weather extremes or human interventions, coupled with a high turnover rate in dry vegetation. The patterns agree with FluxSat and exceed those that FluxCom models estimated."* |
| L477-484: This paragraph is also pretty light on citations. A couple suggestions: Smith et al. 2019 (Remote Sensing of Environment) on challenges specifically in dry regions and the early MODIS papers by Turner that assessed biome differences in MODIS GPP performance. It'd be interested to see the results here contextualized with the challenges that have faced remote sensing of productivity for a long time! | Thanks for suggesting these references! We will provide more references in this paragraph. | We rephrased the paragraph and included the suggested references

l. 585: *"Higher standard errors may indicate that monthly remote sensing and modeled input data are better proxies for some ecosystems than others. For example, GPP can be predicted with low relative uncertainty for ecosystems with a high seasonal variation of biomass, such as croplands, broadleaf forests, and mixed forests. In contrast, predicting GPP in drylands can be more challenging. Drylands are highly sensitive to water availability, resulting in abrupt responses to precipitation and drought events (Barnes et al., 2021). They are characterized by high spatial heterogeneity and irregular temporal vegetation patterns, which are difficult to capture at our spatial and temporal resolution. Together with a low vegetation signal-to-noise ratio, these factors pose a considerable challenge for GPP remote sensing (Smith et al., 2019)."* |
| L481: It's unclear what's meant by "high proportion of biomass" or how that would affect productivity estimation. To me, it | We will rephrase this paragraph. | l. 589: *"high seasonal variation"* |

| | | |
|---|---|---|
| seems like it's not high biomass that would lead to good performance but rather high seasonal variation in leaf area (which both DBF and MF have). | | |
| L484: A little unclear what's meant by "complex biophysical and environmental characteristics." I think it'd be worth expanding on this and being more specific. | We will rephrase this paragraph. | We included a more extensive explanation, highlighting spatial heterogeneity, sensitivity to water availability, and irregular temporal vegetation patterns

**l. 587:** *"In contrast, predicting GPP in drylands can be more challenging. Drylands are highly sensitive to water availability, resulting in abrupt responses to precipitation and drought events (Barnes et al., 2021). They are characterized by high spatial heterogeneity and irregular temporal vegetation patterns, which are difficult to capture at our spatial and temporal resolution. Together with a low vegetation signal-to-noise ratio, these factors pose a considerable challenge for GPP remote sensing (Smith et al., 2019). In an attempt to assess the uniqueness of NEE measurements at FLUXNET sites, Haughton et al. (2018) showed that drier sites and shrubland sites had a higher discrepancy between locally and globally fit models and exhibited more idiosyncratic NEE patterns compared to others. Our results show a similar behavior, with higher model uncertainty for GPP in dryland and shrubland regions."* |
| L487: I think "It is to further research to…" is also somewhat awkward and passive phrasing and would suggest rewording. | We will adapt the text. | **l. 597:** *"We suggest further research ways to improve the performance in low-GPP regions."* |
| L490: This is another good place to cite Smith et al. 2019, which also shows that drylands are underrepresented in flux networks relative to their global proportion. Haughton et al. 2018 (Biogeosciences) could be a good one too since they showed that drylands are more | Thank you for providing these references. We will consider them in the text. | We included the references and added a separate paragraph about the 'uniqueness' of dryland sites, see above (**l. 591**). |

| | | |
|---|---|---|
| "unique" (meaning less easy to apply a globally-trained model to an unseen site) than most other systems, which may be partly why the underrepresentation of dryland sites in flux networks can be such a problem for upscaling in those regions. | | |
| L504: For the Conclusions section, it might be worth expanding on what's meant by "RS" here. That's referring to a specific subset of the variables but for readers who are skimming and skip to the conclusions section, they might miss what that subset refers to. | We will adapt the text. | We spelled out the abbreviations |
| L519-520: Maybe to some extent, but it's interesting to note that RF (not automated and with, I think, some amount of subjectivity in choices) performed nearly as well as the AutoML methods. | It is an interesting point. We will include this in the text. | We mentioned the performance of the RF model

**l. 642:** *"In particular, the non-automated Random Forest model performed almost as well as AutoSklean."* |

**Community comment #1**

| Referee comment | Authors' response | Authors' changes |
|---|---|---|
| When comparing estimations derived from "RS" and "RS + meteo", and observing no substantial improvement in model performance with additional meteorological predictors, the assertion that this is because meteorological data contains no additional information or the reanalysis data quality is not good might need further exploration (Lines 435-440). Given that several predictors from "RS + | You are right that the predictors are likely to contain overlapping information at a monthly scale, and thus, the apparent results by comparing "RS" and "RS+meteo" potentially undermines the actual contribution of meteorological factors to GPP prediction. We aimed to interpret this result in the context of the overall model predictive performance measured by goodness-of-fit metrics. | We underscored that the variable importance measure is model-specific and limited in its representation of the intrinsic value of a variable. We furthermore highlighted, that the analyzed marginal enhancement in modeling GPP is limited to the models used in this study.

**l. 512:** *"Including the meteorological explanatory features (ERA5-Land) in the training data does not significantly improve the prediction quality for any of the frameworks. This implies that meteorological data may not contain additional information that* |

| | | |
|---|---|---|
| meteo" might contain overlapping information on a monthly scale (e.g., VIs, LAI, SIF, ET, and meteorological data), it might be premature to conclude that the inclusion of meteorological data yields marginal enhancement in modeling monthly GPP. | Thus, we will adapt the corresponding text and emphasize that the reanalysis data does not additionally improve the predictive accuracy since meteorological data largely contains overlapping information with the RS variables. We will further underscore that metrological conditions are themselves important controls of GPP in the context of literature. | *the machine learning frameworks in this study can effectively use to predict GPP. A possible explanation is that the "RS" set already includes variables, such as LST, ET, and soil moisture, that encode information about the instantaneous environmental stress on LUE due to adverse meteorological conditions, which are important controls of GPP (Bloomfield et al., 2023). At a monthly scale, the information contained in the meteorological data may overlap with the data provided by the "RS" variables. Furthermore, the coarse resolution of the reanalyzed meteorological data could introduce additional uncertainty due to a scale mismatch with the flux tower footprint sizes. Finally, its quality may not adequately inform the machine learning models due to the presence of large uncertainties. For example, Joiner and Yoshida (2020) showed that using site-measured meteorological data rather than reanalyzed data significantly improved the performance of GPP predictions. Further studies could potentially evaluate these uncertainties by comparing models trained with tower meteorological data to gridded reanalysis datasets."* |
| I am puzzled by the decision to leave out radiation (BESS_Rad) in the 'RS meteo' (Figure 3) and curious about the thinking behind splitting data sources into remote sensing and reanalysis, instead of classifying them into physical (BESS_Rad, ESA CCI, MODIS LST, and ERA5-Land) and biological (MODIS VI/LAI, CSIF, and ALEXI ET) controls. Also, I think it would be worthwhile to discuss whether SIF should be included as a predictor since it is commonly used as a GPP proxy. | BESS_Rad is part of the RS meteo variable set, as stated in figure 3 ("Features of RS + ERA-5 Land"). We will clarify this point in the text. Splitting the data into physical and biological controls is an interesting approach and would certainly give another valuable angle at variable importance. However, it is potentially difficult in terms of drawing the boundaries between these categories (since, for instance, LST and soil moisture are significantly influenced by biological controls). It. In this regard, we will perform an additional analysis to assess the feature importance of individual | The BESS_Rad products are part of RS meteo. We concluded that this is sufficiently highlighted in Table 2 and Figure 3 |

| | variables based on a permutation approach. We expect the result to provide a comprehensive quantification of variable importance and the relative contribution of physical and biological controls. | |
|---|---|---|
| While the Discussion does touch on various potential sources of uncertainties (e.g., section 4.2), it seems to overlook the potential for bias inherent in the eddy covariance GPP. The authors used night-time partitioned GPP, relying quite a bit on a temperature dependency function of night-time NEE. But there is still some debate about whether this dependency is exponential (Chen et al., 2023), if it can be extrapolated to the daytime (Keenan et al., 2019), and whether it should be referenced to air or soil temperature (Wohlfahrt & Galvagno, 2017). Given that AutoML isn't the easiest to interpret (Line 330), I am wondering if its top-notch performance is partly because it is picking up on some error structures during NEE partitioning. | Thank you for raising this relevant point. We will explain the origin of the GPP estimates and how they can affect predicion performance/uncertainty better in the text. Thank you also for providing the references, which we will consider in the text. | We added an additional paragraph on the limitations of night-time partitioned GPP to section 4.4

**l. 603:** *"Finally, an additional limitation is introduced by the eddy covariance measurements themselves. We use night-time-partitioned GPP, which is modeled as the difference between NEE and ecosystem respiration. While NEE and night-time respiration are directly measurable, daytime respiration is modeled with a temperature response function, which extrapolates from night-time respiration (Reichstein et al., 2005). Up to this point, it is not conclusively clarified how reliably this approach can be employed, considering that it is indifferent to some environmental stress factors and changes in respiration behavior between day and nighttime (Wohlfahrt and Galvagno, 2017; Keenan et al., 2019; Tramontana et al., 2020). The inherent uncertainty and bias in the ground truth GPP data could be a potential cap to the performance we can obtain in our efforts to predict GPP."* |
| I am excited about a new global GPP product. Would the authors like to give it an official name, and give the name a spotlight in the Title or Abstract? Additionally, it is recommended that the authors articulate both the interannual variability and the annual magnitude of GPP relative to the new product, as such | Our analysis focuses primarily on the benchmark of different AutoML frameworks. The upscaled maps were mainly used to verify the results of AutoML in comparison to benchmarking products. At this stage, the release of a new GPP dataset is not planned but could be considered in the future. 500m RS data | We included an explanation why we didn't use 500m, however, we encourage upscaling efforts at higher resolutions to increase prediction performance and robustness.

**l. 554:** *"However, we found that the computational demands of the higher resolution made the global upscaling difficult. We suggest further exploring how to align the datasets better, e.g.,* |

| | | |
|---|---|---|
| information would likely be invaluable to the flux community. I am also curious about why the authors did not use the high-resolution RS data (500 m) for the product, considering it seems to pull better performance. | did indeed improve the performance significantly; we used the 0.05 degree data for upscaling in order to compare with other upscaled datasets which are typically at a similar or coarser resolution. In light of our result, production of 500m-resolution data from upscaling is highly encouraged to improve accuracy and reduce uncertainties associated with scaling errors. We will clarify and highlight these aspects in the discussion section. | *through better representing the flux tower footprints (Xiao et al., 2008; Yu et al., 2018; Chu et al., 2021)."* |
| Line 90: Since negative outliers are in a unit of "gC m-2 d-1", did the authors aggregate daily values to monthly for both fluxes and their predictors? More details should be provided for the quality control. | We used the monthly data provided by the original data sources, i.e., FLUXNET2015, AmeriFlux ONEFLUX, and ICOS. The monthly data is aggregated from daily and half-hourly/hourly values. Outlier removal were performed on monthly data that corresponds to average daily NEE. We will provide clarification in the text. | **l. 98:** *"We used the GPP derived from NEE using the night-time partitioning approach (Reichstein et al., 2005), and negative GPP outliers were truncated at -1 gC m-2 d-1 average daily GPP."* |
| Add the source/reference for IGBP here, and also in Fig 2. | We will include this in the text. | Included in the text |
| Line 115: It is a very minor point, but I think terminology for explanatory variables/predictor (e.g., Table 1)/feature (e.g., line 40) is used a bit random in the manuscript. Though they share the same meaning, readers might get confused. | We will better align the terminology. | We changed the terminology, referring to variables/features as 'explanatory' only |
| Line 130-140: It might be worthwhile to relocate this paragraph concerning the challenges with CASH to the Introduction to serve as an additional motivation statement. In the current Introduction, | Thank you for raising this point. We will include the challenge of algorithm selection and hyperparameter tuning in current ML-based products in the introduction. | We included an additional statement about the resource-intensiveness of algorithm selection and hyperparameter tuning as motivation for the study |

| | | |
|---|---|---|
| the authors highlighted the advantages of using AutoML, which are "... to overcome the challenges of algorithm selection, hyperparameter tuning, and pipeline creation through an automated approach". They introduced well the existing problem of feature selection. However, the knowledge gaps in the existing ML-based products of fluxes regarding algorithm selection and hyperparameter tuning should also be clarified. | | **l. 66:** *"Navigating the search space created by the choice of model architecture, hyperparameters, and preprocessing steps to find a suitable combination for GPP prediction is a resource-intensive task. Therefore, researchers often evaluate a selection of combinations that they expect to perform well, thereby potentially missing out on the optimal solution (Karmaker et al., 2021)."* |
| Line 255: Offering details about the calculation of trends, seasonality, across-site variability, and anomalies in the Methodology section, prior to Figure 10, might enhance comprehension. I am also unsure what R2 values mean for trend comparison, as trends are the fitted slopes. | We will include a reference to 2.3.2 to clarify the calculation of trends, seasonality, across-site variability, and anomalies. The R2 for trends represents the spatial variability of their slopes. We will clarify that in the text. | We included a reference to 2.3.2 and an explanation how we calculated r2 for trends

**l. 261:** *"We computed the components as follows: we obtained trends by linear regression of the entire time series (using the slope for evaluation with RMSE and r2), seasonality (mean seasonal cycle) by month-wise averaging, and anomalies as their residuals after detrending and removing seasonality. Furthermore, we calculated an across-site variability from the multi-year mean at each site. For this analysis, we considered only sites with a minimum of 24 months of measurements to minimize the error from sites with just a few measurements, leaving us with 211 sites. When calculating trend metrics, we only considered sites with at least 60 months of measurements for our trend evaluations. Time series anomalies were detrended only when this minimum was reached; otherwise, we simply removed the seasonal component from the time series."* |
| Figure 7: what do R2 values smaller than -1 mean? | We define R2 as the coefficient of determination that provides a measure of the proportion of variation that can be predicted from the predictors variables. | We provided a reference to the Nash-Sutcliffe model efficiency

**l. 257:** *"We used the RMSE and the coefficient of determination (r2) to evaluate the frameworks' performance by comparing the* |

| | | |
|---|---|---|
| | Negative R2 values mean that the model performs worse than a simple model that just predicts the mean of the dependent variable. This definition of R2 aligns with the Nash- Nash–Sutcliffe model efficiency coefficient that is typically used in hydrological models, and it is commonly used as a metric for regression models in machine learning applications. We will provide more descriptions in the text to improve clarity. | *out-of-fold predictions to the ground truth values of GPP. The latter aligns with the Nash-Sutcliffe efficiency (Nash and Sutcliffe, 1970) used in some literature as a performance metric for the GPP prediction (e.g., Tramontana et al. (2016))."* |
| Line 490: While the models also underestimate large GPP values (Line 305), further discussion on this aspect may provide additional insight. | We will include further discussion/literature regarding this behavior. | **l. 469:** *"AutoSklearn trained on "RS" explanatory variables tended to overestimate small GPP values while underestimating large GPP 470 values. This behavior was already observed in the FluxCom (RS), FluxSat, and several light use efficiency models (Yuan et al., 2014; Joiner et al., 2018). It has also been shown for the early MODIS GPP product (Running et al., 2004), where the overestimation was attributed to an artificially high FPAR while the underestimation was related to low light use efficiency in the MODIS algorithm (Turner et al., 2006). Another reason could be the strong reliance of the AutoSklearn framework on tree-based models (Fig. 10). These models are constructed by recursively partitioning the feature space into small regions to which they fit a simple model, which limits them in their ability to extrapolate beyond the range of target values already observed. Furthermore, our predictions showed differing prediction quality at the land cover level, which might result from biome-specific circumstances and the availability of measurement sites. For example, biomes with a pronounced seasonal cycle, such as DBF or MF, exhibit high overall r2, whereas EBF and WET show large variability that the model could not capture. In addition, variability within a land cover type could affect the performance assessment, such as for SH, which includes both arid and subarctic shrublands."* |

| | | |
|---|---|---|
| Line 520: I appreciate the authors raising this point about the cautious use of AutoML. The inherently 'black-box' nature of AutoML, which presents challenges in interpretability as indicated (Line 330), is a notable issue. | Thanks for this feedback! | |

---

## Author Response (AR2)

**Point-to-point reply**

**Referee #1**

| Referee comment | Authors' response | Authors' changes |
|---|---|---|
| Define 'feature importance'. Clearly state what you refer to with 'the importance of features' or 'the contribution of variables'. It seems you refer to the contribution to/ the importance for model accuracy, but this only becomes clear after having read the complete manuscript. In several instances it is not clear what is meant, it could also be the contribution to/the importance for the predicted GPP value itself, as opposed to its accuracy. Examples of instances with need for clarification are in paragraph l.49-64, l.210, 415. | Thank you for this suggestion. Yes, we refer indeed to the importance for model accuracy. We will include a clear definition of variable importance in the introduction of the manuscript. Furthermore, we will make sure that the terminology is consistent throughout the manuscript. We assume you are referring to l.241, l.491 and hope that our changes will resolve these clarity issues. | **l.50-53:** "The contribution of different explanatory variables, such as greenness measures, photosynthetically active radiation (PAR), land surface temperature (LST), soil moisture (SM), and meteorological variables (vapor pressure deficit, temperature, precipitation) to the accuracy of the GPP predictions (hereafter referred to as variable importance) has not been conclusively clarified."

**l.245-247:** "The explanatory variable sets can provide information about the importance of the input features on the performance of the upscaling frameworks."

**l.502-503:** "AutoML is a powerful approach for assessing the importance of the variables on model performance since it selects the optimal base models and constructs optimal pipelines independently for each feature set under consideration." |
| Do you perform any quality checks on the MODIS products before aggregating to monthly and 0.05deg? Data quality is another very important factor determining the accuracy of model results, so information whether and if so, how this has been handled in the work is necessary to report in the manuscript. How do you handle data gaps or low sample availability within a month? | We will reformulate the paragraph on quality checks and gap filling (l.134-142) to make it clearer and more understandable. | **l.142-152:** "We filtered the data for poor-quality pixels, performed gap-filling, and matched spatial and temporal resolutions. We used NBAR (MCD43C4 v006), where more than 75 % of high-resolution NBAR pixels were available from the full BRDF inversion. We selected LST data by applying the quality control mask and where the average emissivity error was less than 0.02. LAI and FPAR were used when retrieved using the main algorithm with or without saturation. Data gaps were filled at the native resolution, similar to the procedure of Walther et al. (2022). We filled gaps of less or equal five days (8 days for four-day resolution datasets) with the average of a fifteen-days moving window for high-frequency datasets (NBAR, LAI, FPAR, BESS_Rad, CSIF). We gap-filled LST with a 9-day moving window because we observed |

| | | higher variations. For SM, we used the moving window median for short gaps and the mean seasonal cycle for long gaps. Finally, we resampled all datasets to 0.05 ° spatial resolution and monthly temporal resolution. Coarser-resolution datasets were resampled using a nearest neighbor approach, while high-resolution data was down-sampled using the conservative remapping method (Jones, 1999)." |
|---|---|---|
| I still wonder what potential consequences it has that many of the features included are model output themselves, partly driven by very similar remotely sensed data sets like in the feature set. Any speculations or justification? | The features used for the modeled datasets, such as CSIF and ET, largely overlap with the features used to model GPP in this study. CSIF draws from the MODIS NBAR product whereas ET is using MODIS LST, LAI, albedo, ASTER surface emissivity, land cover (Hansen), and a GTOPO DEM. Despite these overlaps, they incorporate information that is not provided to our model. The CSIF dataset is trained on OCO-2 SIF data, allowing their model to establish a functional relationship specifically between NBAR and SIF. The ET data, on the other hand, is modeled from a process-based model, which includes domain knowledge that may be challenging for our ML approach to learn.
With ideal model capabilities and scalable training data, we would expect a redundancy of these input data since these relationships would be learned inherently by the ML models just from the MODIS input features. However, given that the ML models might not capture all relationships and that they are trained on | **l.134-140:** "Many of the explanatory variables are themselves datasets that have been modeled from MODIS data. For instance, SIF was predicted from MODIS NBAR using a feed-forward neural network, trained on OCO-2 SIF retrievals (Zhang et al., 2018). ET estimates were modeled by a coupled land-surface and atmospheric boundary layer model (Atmosphere Land Exchange Inverse, ALEXI), which used MODIS LST and LAI as inputs, among others (Hain and Anderson, 2017). Although their input data largely overlap with the inputs to our model, we expected additional improvements from including these datasets due to the domain knowledge of their models, which would otherwise be difficult to replicate in this study by solely relying on MODIS data and limited GPP measurements." |

| | limited GPP measurements (difference in scale), the datasets might provide additional information that are useful for the GPP modeling. The variable importance analysis provides further insights. We will also include a statement highlighting this matter. | |
|---|---|---|
| What is the reason for leaving out the vegetation indices from the RS and RSmeteo feature sets? | We will provide more clarity about the use of the VIs in the manuscript. While VIs slightly improved the performance of the RS minimal datasets, we could not detect any performance improvements in the other datasets. However, this is not one of our main findings in this study and might confuse the reader more than provide insights. For better understanding, we will not consider the VI variables as a separate variable set anymore and instead just mention the finding concerning the VIs directly in the text and in an additional table in the appendix. | **Tab. 1,2:** Removed the VIs
**Fig. 3-6:** Removed RS minimal +VI

**l.232-241:** ” We organized the explanatory variables into three sets to determine their impact on GPP predictions within different AutoML frameworks (Tramontana et al., 2016; Joiner and Yoshida, 2020). Each set consisted of different features that could explain the variation in GPP. The minimal set of remotely sensed variables (RS minimal) included surface reflectance from seven MODIS visible to infrared bands and PAR, which largely reflect the ability of the vegetation canopy to intercept solar radiation for photosynthesis. The "RS" set included all remotely sensed variables and their products. Notably, compared to the "RS minimal" set, the "RS" set also included land surface temperature, evapotranspiration, and soil moisture, which provide an additional link to vegetation heat and water stress (Green et al., 2022; Stocker et al., 2018). Finally, the "RS meteo" set included all remotely sensed variables and, in addition, meteorological variables from the ERA5-Land reanalysis (see Table 2). Additionally, we replaced the MODIS reflectance bands, LAI, FPAR, and land cover products with their native 500 m resolution data in the "RS" set to evaluate the impact of satellite data spatial resolution on GPP estimation.”

**l.346-347:** “In addition, we evaluated whether vegetation indices (VI) could improve the performance of the variable sets, but no |

| | | improvements were found beyond the "RS minimal" dataset (Tab. A1)." |
|---|---|---|
| | | **Tab. A1:** Added an additional table of mean r2 values to the appendix for the repeated cross-validations, including the results with VIs. |
| Please clarify which data product versions were used for the ESA CCI soil moisture and for the Fluxcom (l.250 states Fluxcom v6, so does this refer to the Fluxcom RSonly data from MODIS c006?). | We used ESA CCI v06.1 (see Table 1) and FluxCom v6, RS only (see l.284). This refers to RS only from MODIS collection 006. We will highlight this in the text. | **l.287-289:** "We produced global GPP and standard error maps at a resolution of 0.05 ° in monthly frequency from 2001 to 2020, which we compared with the two ML-based reference datasets FluxCom v6 (RS only, based on data from the MODIS collection 6) (Jung et al., 2020) and FluxSat (Joiner and Yoshida, 2020)." |
| How was the R2 computed (http://www.jstor.org/stable/2683704)? | Thank you for raising this matter. Our R2 definition aligns with the Nash-Sutcliffe model efficiency (l.258), which corresponds to Eq. 1 in Kvalseth (1985). We will include an equation in the appendix. | **Eq. A1:** $$r^2 = 1 - \frac{\sum_{i=1}^{N}(y_i - \hat{y}_i)^2}{\sum_{i=1}^{N}(y_i - \bar{y})^2}$$ |
| Figure 4 and 5: Do these distributions represent 30 (cross-validation rounds) R2 values computed across all sites, or 30 (cross-validation rounds) x 245 (sites) R2 values computed for each site and shown all together? For spatial variability I understand it is always 30, right? Similarly, the question on the grouping for the R2 in Fig.7. | Thank you for raising this question. The graphs show the distribution of the R2-values from the 30 repeated cross-validations (hence, a distribution of 30 R2-values, the first part of your question). Within each cross-validation, the R2 is calculated over the entire prediction, in which all sites are merged. This applies to total, trend, seasonality, anomalies, and also spatial variability. The only difference for spatial variability is, that we compare averages instead of temporal time series components. We will formulate the corresponding text passages more clear. | **l.302-304 (Caption fig. 4):** "Overall framework performance, expressed as the coefficient of determination ($r^2$) for the candidate frameworks and the three different explanatory variable sets. Each distribution belongs to one framework and one set of explanatory variables and results from the repeated cross-validations, for each of which one $r^2$ value is calculated over the predictions at all sites."

 **l.313-317 (Caption fig. 5):** "Evaluation of the temporally and spatially decomposed time series expressed as the coefficient of determination ($r^2$). Each distribution belongs to one framework and one set of explanatory variables and results from the repeated cross-validations, for each of which one $r^2$ value is calculated over the predictions at all sites. The $r^2$ values for seasonality and anomalies were calculated from seasonal cycles |

| | | and anomalies at monthly granularity, while those for trend and across-site variability were calculated from one trend or mean value per site, respectively." |
|---|---|---|
| | | **l.365-366 (Caption fig. 7):** "The distribution results from the repeated cross-validations, for each of which one $r^2$ value is calculated over the predictions at all sites." |
| l. 485-490: Would you expect more (dummy) training data or feature selection enabled to be more promising for higher model accuracy? | We expect higher robustness of the model predictions and the evaluation metrics from more and better-balanced training data. | **l.499-500:** "More training data with better geographic representation could help mitigate these shortcomings and could lead to more robust predictions, model evaluations, and potentially higher model performance." |
| l.14/15: These are some, but not all choices that affect the accuracy of the regression model. | We will highlight that these are just some aspects. | **l.13-15:** "However, the accuracy of the regression model can be affected by uncertainties introduced by model selection, parameterization, and choice of explanatory features, among others." |
| Discussion on light-use-efficiency is unclear to me given the monthly temporal scale of interest in this study. Line 428, 436 contrast instantaneous GPP reductions due to environmental stress. I suggest to rephrase this paragraph and give the (in my opinion) more reasonable explanation of the difference between site level and reanalysis meteorology more visibility. | Thank you for this suggestion. We are a bit unclear as of to what you refer to by l. 428 and l. 436. We assume you are referring to l. 507 and l. 515 and hope that our changes satisfactorily address your concerns. We will reformulate this paragraph to highlight the scale mismatch better as a possible explanation. | **l.522-532:** "Including the meteorological explanatory features (ERA5-Land) in the training data does not significantly improve the prediction quality for any of the frameworks. This implies that meteorological data may not contain additional information that the machine learning frameworks in this study can effectively use to predict GPP. A possible explanation could be the mismatch between reanalysis and site meteorology. The coarse resolution and large uncertainties of the reanalysis data may result in a poor representation of the flux tower footprints, which are often smaller than one pixel of the reanalysis data, leading to uncertainties in the modeling. For example, Joiner and Yoshida (2020) showed that using site-measured meteorological data instead of reanalyzed data significantly improved the performance of GPP predictions. At the monthly scale, the "RS" variable set may already encode information about the instantaneous environmental stress from adverse meteorological conditions through, for example, LST, ET, and soil moisture, which are important controls on GPP (Bloomfield et al., 2023). Further |

| | | studies could potentially assess these uncertainties by comparing models trained with tower meteorological data to gridded reanalysis datasets." |
|---|---|---|
| Discussion of spatial resolution l. 441-445: To me this paragraph suggests between the lines that training such models by pairing eddy-covariance data with 0.05 or 0.25 pixels as done in this work is the state-of-the-art. It is not. And this could become more clear. For some variables there is no other option because data are not available at finer spatial resolution, this is clear. But the authors chose to take the coarser pixels as the normal standard, and I suggest to make this difference between author choice and state-of-the-art clear. | We assume you refer to l.551-556. We will highlight that the spatial resolution in this study is a result of the authors' choice and reformulate the paragraph less suggestive. | l.563-565: "These results underscore the importance of spatial resolution and suggest the use of data with a resolution that better represents smaller landscape features and flux tower footprints, in contrast to our initial choice of 0.05 ° resolution in this study. (Xiao et al., 2008; Yu et al., 2018; Chu et al., 2021)." |

**Referee #2**

| Referee comment | Authors' response | Authors' changes |
|---|---|---|
| Line 129: I think these variables could use a little more explanation plus citations. My reasoning here is, as I mentioned in my previous comments, that several of these variables (SIF and ET) are themselves modeled from remote sensing data, and I think that is important context for how they are interpreted. The SIF product used here, for example, is not truly a measured SIF signal but a modeled SIF based solely on surface reflectance (the NBAR product). In my opinion, it's important to | We will include citations and refer to the variables' modeled background. SIF is modeled from MODIS NBAR and OCO-2 SIF retrievals, the former of which is also an input to our GPP models. ET is modeled from MODIS day and nighttime LST, MODIS LAI, MODIS albedo, ASTER surface emissivity, land cover (Hansen), and a GTOPO DEM, hence overlapping in terms of LAI and LST. We will provide more context on these variables. | l.134-140: "Many of the explanatory variables are themselves datasets that have been modeled from MODIS data. For instance, SIF was predicted from MODIS NBAR using a feed-forward neural network, trained on OCO-2 SIF retrievals (Zhang et al., 2018). ET estimates were modeled by a coupled land-surface and atmospheric boundary layer model (Atmosphere Land Exchange Inverse, ALEXI), which used MODIS LST and LAI as inputs, among others (Hain and Anderson, 2017). Although their input data largely overlaps with the inputs to our model, we expected additional improvements from including these datasets due to the domain knowledge of their models, which would otherwise be difficult to |

| | | |
|---|---|---|
| make that clearer here since I definitely think it affects the interpretation of its importance as a variable. | | replicate in this study by solely relying on MODIS data and limited GPP measurements." |
| Line 137: I think just a little more detail about the resampling could be helpful. For products with finer resolutions than 0.05 degrees, were all pixels within the 0.05 degree cell averaged together? For those with coarser resolutions, how were they down-scaled to 0.05 degrees? | We performed the down sampling using the conservative remapping method, and the up sampling using nearest neighbor. We will include this in the manuscript. | **l.149-152:** "Finally, we resampled all datasets to 0.05 ° spatial resolution and monthly temporal resolution. Coarser-resolution datasets were resampled using a nearest neighbor approach, while high-resolution data was down-sampled using the conservative remapping method (Jones, 1999)." |
| Lines 331-332, 501-502, and 626-628: I think this is slightly overstating the improvement of the RS set over the RS-minimal and RS-minimal+VI sets. Per 331-332, the full RS set only added 2% variance explained, so I think it's too strong to say that the NBAR + PAR "did not provide the models with sufficient information." I think it would be more accurate to say that NBAR + PAR is responsible for the vast majority of model skill but the remaining variables can add some additional information on the margins. | Thank you for this valid point. We will rephrase the corresponding paragraph and make it less overstating. | **l.511-520:** "The frameworks' performance depends significantly on the choice of predictive features on which they are trained. The results show that while the seven NBAR bands and PAR from the "RS minimal" variable set provide the model with sufficient information for a GPP prediction, the full set of "RS" variables adds additional information that all the frameworks can exploit. The additional variables in the "RS" variable set, such as SIF, LAI, FPAR, ET, LST, SM, and plant function type, appear to include important environmental forcings and structural variables that provide a marginal advantage over the variables on only vegetation structure and radiation in "RS minimal" (Green et al., 2019; Stocker et al., 2019; Xu et al., 2020)."

**l.633-637:** "We found that remotely sensed (RS) explanatory variables provided the best results in combination with the investigated frameworks. While only relying on the MODIS NBAR reflectance bands and PAR ("RS minimal") provided the models with sufficient information for GPP prediction, considering other proxies of photosynthetic activity and canopy structure, such as solar-induced fluorescence, leaf area index, and fraction of absorbed photosynthetic activity, increased the performance of all models." |

**Community comment #1**

| Referee comment | Authors' response | Authors' changes |
|---|---|---|
| I have carefully examined the authors' responses to the previous comments and the changes made in the revised manuscript. The authors have thoughtfully addressed the concerns raised, enhancing the quality of the work. I congratulate the authors on their diligent efforts and appreciate the opportunity to review this manuscript. Based on the revisions made, I recommend the manuscript be accepted for publication. | Thank you for your review. We appreciated your suggestions, which have greatly improved this manuscript. | |